# FORMAL VERIFICATION FOR NEURAL NETWORKS WITH GENERAL NONLINEARITIES VIA BRANCH-AND-BOUND

## ABSTRACT

Bound propagation with branch-and-bound (BaB) is so far among the most effective methods for neural network (NN) verification. However, existing works with BaB have mostly focused on NNs with piecewise linear activations, especially ReLU networks. In this paper, we develop a framework for conducting BaB based on bound propagation with general branching points and an arbitrary number of branches, as an important move for extending NN verification to models with various nonlinearities beyond ReLU. Our framework strengthens verification for common neural networks with *element-wise* activation functions, as well as other *multi-dimensional* nonlinear operations such as multiplication. In addition, we find that existing heuristics for choosing neurons to branch for ReLU networks are insufficient for general nonlinearities, and we design a new heuristic named BBPS, which usually outperforms the heuristic obtained by directly extending the existing ones originally developed for ReLU networks. We empirically demonstrate the effectiveness of our BaB framework on verifying a wide range of NNs, including networks with Sigmoid, Tanh, sine or GeLU activations, LSTMs and ViTs, which have various nonlinearities. Our framework also enables applications with models beyond neural networks, such as models for AC Optimal Power Flow (ACOPF).

## 1 INTRODUCTION

Neural network (NN) verification aims to formally verify whether a neural network satisfies specific properties, such as safety or robustness, prior to its deployment in safety-critical applications. Mathematically, verifiers typically compute bounds on the output neurons within a pre-defined input region. As computing exact bounds is NP-complete (Katz et al., 2017) even for simple ReLU networks, it becomes crucial to relax the bound computation process to improve efficiency. Bound propagation methods (Wang et al., 2018b; Wong & Kolter, 2018; Zhang et al., 2018; Dvijotham et al., 2018; Henriksen & Lomuscio, 2020; Singh et al., 2019b) are commonly used, which relax nonlinearities in neural networks into linear lower and upper bounds that can be efficiently propagated. Linear relaxation relies on intermediate layer bounds, which are recursively computed with bound propagation. However, if intermediate bounds are not sufficiently tight, relaxation often results in loose output bounds, particularly for deeper networks.

To further tighten the bounds for bound propagation, Branch-and-Bound (BaB) has been widely utilized (Bunel et al., 2018; 2020; Xu et al., 2021; Lu & Mudigonda, 2020; De Palma et al., 2021; Wang et al., 2021; Ferrari et al., 2021). BaB iteratively branches intermediate bounds so that the original verification is branched into subdomains with tighter intermediate bounds. Subsequently, these subdomains can be bounded individually with tighter linear relaxations. However, previous works mostly focused on ReLU networks due to the simplicity of ReLU from its piecewise linear nature. Branching a ReLU neuron only requires branching at 0, and it immediately becomes linear in either branch around 0. Conversely, handling neural networks with nonlinearities beyond ReLU, such as LSTMs (Hochreiter & Schmidhuber, 1997) and Transformers (Vaswani et al., 2017) which also have nonlinearities beyond activation functions such as multiplication and division, introduces additional complexity as the convenience of piecewise linearity diminishes. There have been previous works consdering BaB for NNs beyond ReLU networks, e.g., Henriksen & Lomuscio (2020); Wu et al. (2022) considered BaB on networks with S-shaped activations such as Sigmoid. However,

these works still often specialize in specific and relatively simple types of nonlinearities, and a more principled framework for handling general nonlinearities is lacking, leaving ample room for further advancements in verifying non-ReLU networks.

In this paper, we propose a principled verification framework with BaB for neural networks with general nonlinearities. We generalize $\alpha,\beta$-CROWN[1] (Zhang et al., 2018; Xu et al., 2020; 2021; Wang et al., 2021) which is based on linear bound propagation and BaB. While $\alpha,\beta$-CROWN does accept non-ReLU activations, their BaB is still restricted to ReLU. We resolve multiple challenges to enable BaB for general nonlinearities beyond piecewise linear ReLU. We first formulate a general BaB framework. This formulation encompasses general branching points (in contrast to simply 0 for ReLU) and a general number of branches (in contrast to two branches for ReLU which naturally has two pieces). We also formulate and encode general branching constraints in linear bound propagation by optimizable Lagrange multipliers to tighten the bounds. Moreover, we find that a popular existing branching heuristic named "BaBSR" for selecting ReLU neurons to branch (Bunel et al., 2020) is suboptimal when directly extended to networks with general nonlinearities. It is because for their convenience and efficiency, BaBSR discards an important term which is found to be negligible on ReLU networks, yet we find it to be important on general nonlinearities. Thereby, to improve the effectiveness of BaB, we introduce a new branching heuristic named "Branching via Bound Propagation with Shortcuts (BBPS)" with a more accurate estimation by carefully leveraging the linear bounds from bound propagation.

We demonstrate the effectiveness of the new framework on a variety of networks, including feed-forward networks with Sigmoid, Tanh, sine, or GeLU activations, LSTMs, Vision Transformers (ViTs). We also enable verification on models for the AC Optimal Power Flow (ACOPF) application, which contains a general computational graph beyond a neural network. These models involve various nonlinearities including S-shaped activations, periodic trigonometric functions, and also multiplication and division which are multi-dimensional nonlinear operations beyond activation functions. Our BaB is generally effective and outperforms the existing baselines.

## 2 BACKGROUND

**The NN verification problem.** Let $f : \mathbb{R}^d \mapsto \mathbb{R}^K$ be a neural network taking input $\mathbf{x} \in \mathbb{R}^d$ and outputting $f(\mathbf{x}) \in \mathbb{R}^K$. Suppose $\mathcal{C}$ is the input region to be verified, and $s : \mathbb{R}^K \mapsto \mathbb{R}$ is an output specification function, $h : \mathbb{R}^d \mapsto \mathbb{R}$ is the function that combines the NN and the output specification as $h(\mathbf{x}) = s(f(\mathbf{x}))$. NN verification can typically be formulated as verifying if $h(\mathbf{x}) > 0, \forall \mathbf{x} \in \mathcal{C}$ provably holds. A commonly adopted special case is robustness verification given a small input region, where $f(\mathbf{x})$ is a $K$-way classifier and $h(\mathbf{x}) := \min_{i \neq c}\{f_c(\mathbf{x}) - f_i(\mathbf{x})\}$ checks the worst-case margin between the ground-truth class $c$ and any other class $i$. The input region is often taken as a small $\ell_\infty$-ball with radius $\epsilon$ around a data point $\mathbf{x}_0$, i.e., $\mathcal{C} := \{\mathbf{x} \mid \|\mathbf{x} - \mathbf{x}_0\|_\infty \leq \epsilon\}$. This a succinct and useful problem for provably verifying the robustness properties of a model and also benchmarking NN verifiers, although there are other NN verification problems beyond robustness. We also mainly focus on this setting for its simplicity following prior works.

**Linear bound propagation**. We develop our new framework based on $\alpha,\beta$-CROWN (Xu et al., 2020; 2021; Wang et al., 2021) that is among the state-of-the-art NN verifiers (Bak et al., 2021; Müller et al., 2022a). $\alpha,\beta$-CROWN is based on linear bound propagation (Zhang et al., 2018) which can lower bound $h(\mathbf{x})$ by propagating linear bounds w.r.t. the output of one or more intermediate layers as

$$h(\mathbf{x}) \geq \sum_i \mathbf{A}_i \hat{\mathbf{x}}_i + \mathbf{c}, \tag{1}$$

where $\hat{\mathbf{x}}_i$ $(i \leq n)$ is the output of intermediate layer $i$ in the network with $n$ layers, $\mathbf{A}_i$ are the coefficients w.r.t. layer $i$, and $\mathbf{c}$ is a bias term. In the beginning, the linear bound is simply $h(\mathbf{x}) \geq \mathbf{I} \cdot h(\mathbf{x}) + \mathbf{0}$ which is actually an equality. In the bound propagation, $\mathbf{A}_i \hat{\mathbf{x}}_i$ in Eq. (1) is recursively substituted by the linear bound of $\hat{\mathbf{x}}_i$ w.r.t its input. For simplicity, suppose layer $i - 1$ is the input to layer $i$ and $\hat{\mathbf{x}}_i = h_i(\hat{\mathbf{x}}_{i-1})$, where $h_i(\cdot)$ is the computation for layer $i$. And suppose we have the linear bounds of $\hat{\mathbf{x}}_i$ w.r.t its input $\hat{\mathbf{x}}_{i-1}$ as:

$$\underline{\mathbf{a}}_i \hat{\mathbf{x}}_{i-1} + \underline{\mathbf{b}}_i \leq \hat{\mathbf{x}}_i = h_i(\hat{\mathbf{x}}_{i-1}) \leq \overline{\mathbf{a}}_i \hat{\mathbf{x}}_{i-1} + \overline{\mathbf{b}}_i, \tag{2}$$

---

[1] $\alpha,\beta$-CROWN mentioned in this paper is the version released by March 2023 at `https://github.com/Verified-Intelligence/alpha-beta-CROWN`.

with parameters $\underline{\mathbf{a}}_i, \underline{\mathbf{b}}_i, \overline{\mathbf{a}}_i, \overline{\mathbf{b}}_i$ for the linear bounds, and "$\leq$" holds elementwise. Then $\mathbf{A}_i \hat{\mathbf{x}}_i$ can be substituted and lower bounded by:

$$\mathbf{A}_i \hat{\mathbf{x}}_i \geq \mathbf{A}_{i-1} \hat{\mathbf{x}}_{i-1} + \left( \mathbf{A}_{i,+} \underline{\mathbf{b}}_i + \mathbf{A}_{i,-} \overline{\mathbf{b}}_i \right), \quad \text{where } \mathbf{A}_{i-1} = \left( \mathbf{A}_{i,+} \underline{\mathbf{a}}_i + \mathbf{A}_{i,-} \overline{\mathbf{a}}_i \right), \qquad (3)$$

where "+" and "-" in the subscripts denote taking positive and negative elements respectively, and in this way the linear bounds are propagated from layer $i$ to layer $i - 1$. Ultimately the linear bounds can be propagated to the input of the network $\mathbf{x}$ as $h(\mathbf{x}) \geq \mathbf{A}_0 \mathbf{x} + \mathbf{c}$, $\mathbf{A}_0 \in \mathbb{R}^{1 \times d}$, where the input can be viewed as the 0-th layer. Depending on $\mathcal{C}$, this linear bound can be concretized into a lower bound without $\mathbf{x}$. If $\mathcal{C}$ is an $\ell_\infty$ ball, we have

$$\forall \|\mathbf{x} - \mathbf{x}_0\|_\infty \leq \epsilon, \quad \mathbf{A}_0 \mathbf{x} + \mathbf{c} \geq \mathbf{A}_0 \mathbf{x}_0 - \epsilon \|\mathbf{A}_0\|_1 + \mathbf{c}. \qquad (4)$$

To construct Eq. (2), if $h_i(\cdot)$ is inherently linear. Otherwise, linear relaxation is used, which relaxes a nonlinearity and bound the nonlinearity by linear functions. An intermediate bound on $\hat{\mathbf{x}}_{i-1}$ as $\mathbf{l}_{i-1} \leq \hat{\mathbf{x}}_{i-1} \leq \mathbf{u}_{i-1}$ is usually required for the relaxation, which can be obtained by running additional bound propagation and treating the intermediate layers as the output of a network. Linear relaxation can contain optimizable parameters to tighten the bounds (Lyu et al., 2020; Xu et al., 2021). And we use $\boldsymbol{\alpha}$ to denote all the optimizable parameters in the linear relaxation.

**Branch-and-Bound (BaB).** BaB has been widely applied to tighten verification bounds. Each time it *branches* the intermediate bound of a selected neuron $j$ in a selected layer $i - 1$, $\hat{\mathbf{x}}_{i-1,j} \in [\mathbf{l}_{i-1,j}, \mathbf{u}_{i-1,j}]$, into smaller subdomains with tighter intermediate bounds. Then BaB *bounds* such subdomain respectively and take the worst bound from the subdomains as the new bound. This process is repeated iteratively to gradually improve the bounds. $\alpha,\beta$-CROWN also adds branching constraints derived from the new intermediate bounds after each branching, and the constraints are utilized in bound propagation to tighten the bounds with Lagrangian multipliers. We use $\boldsymbol{\beta}$ to denote all the Lagrangian multipliers. Note that BaB in $\alpha,\beta$-CROWN is restricted to ReLU activation only.

## 3 METHOD

### 3.1 OVERALL FRAMEWORK

In this section, we describe the overall framework, which mostly follows $\alpha,\beta$-CROWN (Zhang et al., 2018; Xu et al., 2020; 2021; Wang et al., 2021). Compared to the original $\alpha,\beta$-CROWN which only supports ReLU neurons in the BaB, we will formulate a general branching framework for general nonlinearities and also a new branching heuristic, in the remaining subsections of Section 3.

**Notations.** In Section 2, we only considered a feedforward NN for simplicity. But the linear bound propagation technique has been generalized to general computational graphs to support various NN architectures (Xu et al., 2020). In our method, we also consider a general computational graph $h(\mathbf{x})$ for input region $\mathbf{x} \in \mathcal{C}$. Instead of a feedforward network with $n$ *layers* in Section 2, we consider a computational graph with $n$ *nodes*, where each node $i$ computes some function $h_i(\cdot)$ that may either correspond to a linear layer in the NN or a nonlinearity. We use $\hat{\mathbf{x}}_i$ to denote the output of node $i$ which may contain many neurons, and we use $\hat{\mathbf{x}}_{i,j}$ to denote the output of the $j$-th neuron in node $i$. Intermediate bounds of node $i$ may be needed to relax and bound $h_i(\cdot)$, and we use $\mathbf{l}_{i,j}, \mathbf{u}_{i,j}$ to denote the intermediate lower bound and upper bound respectively. We use $\mathbf{l}$ and $\mathbf{u}$ to denote all the intermediate lower bounds and upper bounds respectively for the entire computational graph.

**Initial verification.** Before entering BaB, we first compute initial verified bounds by bound propagation with optimizable linear relaxation. Specifically, we use $V_\alpha(h, \mathcal{C}, \boldsymbol{\alpha})$ to denote the linear bound propagation-based verifier with $\boldsymbol{\alpha}$ denoting all the parameters in the optimizable relaxation, and we compute initial verified bounds by optimizing $\boldsymbol{\alpha}$, as $h(\mathbf{x}) \geq \max_{\boldsymbol{\alpha}} V_\alpha(h, \mathcal{C}, \boldsymbol{\alpha})$ ($\forall \mathbf{x} \in \mathcal{C}$), where $\boldsymbol{\alpha}$ is constrained within a domain that ensures the soundness of the relaxation. All the intermediate bounds are also updated with the updating $\boldsymbol{\alpha}$, and we obtain the optimized intermediate bounds $\mathbf{l}, \mathbf{u}$. The verification finishes if $V_\alpha(h, \mathcal{C}, \boldsymbol{\alpha}) > 0$ holds already. Since $\alpha,\beta$-CROWN has limited support on nonlinearities beyond ReLU, we have derived new optimizable linear relaxation we encounter, as discussed in Appendix B.

**Branch-and-Bound.** Otherwise, we enter our BaB to tighten the bounds. We maintain a dynamic pool of intermediate bound domains, $\mathcal{D} = \{(\mathbf{l}^{(i)}, \mathbf{u}^{(i)})\}_{i=1}^m$, where $m = |\mathcal{D}|$ is the number of current

domains, and initially $\mathcal{D} = \{(\mathbf{l}, \mathbf{u})\}$ with the intermediate bounds from the initial verification. In each iteration of BaB, we pick a domain that has the worst verified bounds. For this domain, we select a neuron to branch and obtain new subdomains. For the new subdomains, we update $\mathbf{l}, \mathbf{u}$ for the branched neurons, and we also use $\boldsymbol{\beta}$ parameters for the Lagrange multipliers in the branching constraints. For each new subdomain, given updated $\mathbf{l}, \mathbf{u}$ and the parameters $\boldsymbol{\alpha}, \boldsymbol{\beta}$, we denote a verified lower bound computed during BaB as $V(h, \mathbf{l}, \mathbf{u}, \boldsymbol{\alpha}, \boldsymbol{\beta})$, and we optimize $\boldsymbol{\alpha}$ and $\boldsymbol{\beta}$ to obtain an optimized lower bound for $h(\mathbf{x})$:

$$h(\mathbf{x}) \geq \max_{\boldsymbol{\alpha}, \boldsymbol{\beta}} V(h, \mathbf{l}, \mathbf{u}, \boldsymbol{\alpha}, \boldsymbol{\beta}), \;\; \forall \mathbf{x} \in \mathcal{C}. \tag{5}$$

Subdomains with $V(h, \mathbf{l}, \mathbf{u}, \boldsymbol{\alpha}, \boldsymbol{\beta}) > 0$ are verified and discarded, otherwise they are added to $\mathcal{D}$ for further branching. We repeat the process until no domain is left in $\mathcal{D}$ and the verification succeeds, or when the timeout is reached and the verification fails. We illustrate the framework in Appendix A.

## 3.2 BRANCHING FOR GENERAL NONLINEARITIES

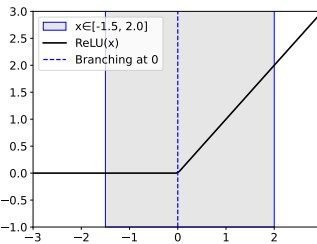
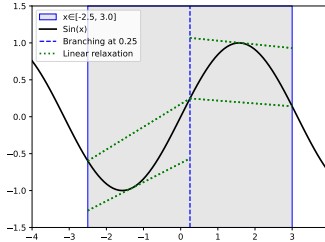
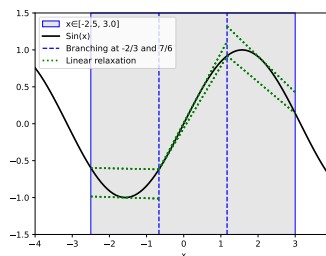

(a) Branching a ReLU activation.

(b) Branching a Sin activation into two branches.

(c) Branching a Sin activation into three branches.

Figure 1: Illustration of branching the intermediate bound of a neuron with different activations. In Figure 1b and 1c, the function is still nonlinear after branching, and we also show the linear relaxation of different branches.

Branching on ReLU networks as studied by prior works is a special case of branching on general nonlinearities. For ReLU networks, branching is needed only if $\mathbf{l}_{i,j} < 0 < \mathbf{u}_{i,j}$ for a neuron, and the only reasonable way is to branch at 0 and split the intermediate bounds into two branches, $[\mathbf{l}_{i,j}, 0]$ and $[0, \mathbf{u}_{i,j}]$, so that ReLU is linear for both sides, as shown in Figure 1a. However, branching for general nonlinearities on general computational graphs is more complex. First, branching can be needed even if $\mathbf{u}_{i,j} \leq 0$ or $\mathbf{l}_{i,j} \geq 0$ and it requires considering branching at points other than 0. Second, unlike ReLU, general nonlinearities usually do not consist of two linear pieces, and the intermediate bounds may be branched into more than two branches at once for tighter linear relaxation (Figure 1b v.s. Figure 1c). Third, unlike typical activation functions, some nonlinearities may take more than one input. For example, there may be a node computing $\hat{\mathbf{x}}_i = h_i(\hat{\mathbf{x}}_{i-1}, \hat{\mathbf{x}}_{i-2}) = \hat{\mathbf{x}}_{i-1}\hat{\mathbf{x}}_{i-2}$, as appeared in Transformers (Vaswani et al., 2017; Shi et al., 2019) or LSTMs (Hochreiter & Schmidhuber, 1997; Ko et al., 2019). The multiplication between $\hat{\mathbf{x}}_{i-1}$ and $\hat{\mathbf{x}}_{i-2}$ is generally a nonlinear function unless one of $\hat{\mathbf{x}}_{i-1}$ and $\hat{\mathbf{x}}_{i-2}$ is constant and does not depend on $\mathbf{x}$. For such nonlinearities, there are multiple input nodes that can be branched. Fourth, on general computational graphs, a node can also be followed by multiple nonlinearities, as appeared in LSTMs, and then branching intermediate bounds of this node can affect multiple nonlinearities.

To resolve these challenges, we propose a new and more general formulation for branching on general nonlinearities for general computational graphs. Each time we consider branching the intermediate bounds of a neuron $j$ in a node $i$, namely $[\mathbf{l}_{i,j}, \mathbf{u}_{i,j}]$, if node $i$ is the input of some nonlinearity. We consider branching the concerned neuron into $K$ branches with branching points $\mathbf{p}_{i,j}^{(1)}, \cdots, \mathbf{p}_{i,j}^{(K-1)}$, and then the intermediate bounds become:

$$[\mathbf{l}_{i,j}, \mathbf{u}_{i,j}] \rightarrow [\mathbf{l}_{i,j}, \mathbf{p}_{i,j}^{(1)}], [\mathbf{p}_{i,j}^{(1)}, \mathbf{p}_{i,j}^{(2)}], \cdots, [\mathbf{p}_{i,j}^{(K-1)}, \mathbf{u}_{i,j}], \tag{6}$$

for the $K$ branches respectively. In this work, we instantiate Eq. (6) as uniformly branching $[\mathbf{l}_{i,j}, \mathbf{u}_{i,j}]$ into $K$ branches where we mainly take $K = 3$ for non-ReLU models. We study the impact of different $K$ values in Appendix C.3.

We select the neuron to branch by a heuristic to approximately maximize the bound improvement after the branching, as discussed in Section 3.4. If neuron $j$ in node $i$ is selected, we use the new intermediate bounds of each branch to update the linear relaxation of the impacted nonlinearities. We also add branching constraints parameterized by $\boldsymbol{\beta}$, as will be discussed in Section 3.3. Then compute new verified bounds for the branches by solving Eq. (5) with multiple iterations optimizing $\boldsymbol{\alpha}$ and $\boldsymbol{\beta}$.

Note that in our formulation, we consider each node that is *the input to some nonlinearities* and decide if we branch on this node, and it allows us to naturally generalizes to nonlinearities with multiple input nodes as well as multiple nonlinearities sharing the the input node. It would be more convenient and general compared to considering *the nonlinearities themselves*, and how all the input nodes of a nonlinearity shall be branched, yet the input nodes may be shared by some other nonlinearities.

### 3.3 General Branching Constraints

We formulate and encode general branching constraints into the linear bound propagation by $\boldsymbol{\beta}$ Lagrange multipliers which have shown to be important for linear bound propagation in BaB (Wang et al., 2021) which focused on ReLU. For each neuron $j$ in a node $i$ branched as Eq. (6), we formulate branching constraints for the output $\hat{\mathbf{x}}_{i,j}^{(1)}, \cdots, \hat{\mathbf{x}}_{i,j}^{(K)}$ in the $K$ branches respectively:

$$\hat{\mathbf{x}}_{i,j}^{(1)} - \mathbf{p}_{i,j}^{(1)} \leq 0, \quad \hat{\mathbf{x}}_{i,j}^{(2)} - \mathbf{p}_{i,j}^{(2)} \leq 0, \, \mathbf{p}_{i,j}^{(1)} - \hat{\mathbf{x}}_{i,j}^{(2)} \leq 0, \quad \cdots, \quad \mathbf{p}_{i,j}^{(K-1)} - \hat{\mathbf{x}}_{i,j}^{(K)} \leq 0. \quad (7)$$

Zhang et al. (2022) has proposed to encode general cutting plane constraints into linear bound propagation to tighten the bounds. Our general branching constraints can also be viewed as a particular type of cutting plane constraints. We add $\mathbf{s}_{i,j}^{(k)}$ for the $k$ ($1 \leq k \leq K$)-th branch:

$$\mathbf{s}_{i,j}^{(1)} \coloneqq \beta_{i,j}^{(1)}(\hat{\mathbf{x}}_{i,j}^{(1)} - \mathbf{p}_{i,j}^{(1)}), \; \mathbf{s}_{i,j}^{(K)} \coloneqq \beta_{i,j}^{(K)}(\mathbf{p}_{i,j}^{(K-1)} - \hat{\mathbf{x}}_{i,j}^{(K)}), \quad (8)$$

$$\mathbf{s}_{i,j}^{(k)} \coloneqq \beta_{i,j}^{(k,1)}(\hat{\mathbf{x}}_{i,j}^{(k)} - \mathbf{p}_{i,j}^{(k)}) + \beta_{i,j}^{(k,2)}(\mathbf{p}_{i,j}^{(k-1)} - \hat{\mathbf{x}}_{i,j}^{(k)}) \quad \text{for } 2 \leq k \leq K-1, \quad (9)$$

which can be added to the right-hand-side of Eq. (1) as $h(\mathbf{x}) \geq \sum_i (\mathbf{A}_i \hat{\mathbf{x}}_i + \sum_j \mathbf{s}_{i,j}) + \mathbf{c}$, where $\beta(1)_{i,j}, \beta_{i,j}^{(K)}, \beta_{i,j}^{(k,1)}, \beta_{i,j}^{(k,2)} \geq 0$ ($2 \leq k \leq K-1$) are Lagrangian multipliers. Compared to Zhang et al. (2022) which focused on utilizing general cutting plane constraints in linear bound propagation, our new contribution here is on formulating general branching constraints which can then be handled in a similar way as Zhang et al. (2022).

### 3.4 A New Branching Heuristic for General Nonlinear Functions

In each branching iteration, we aim to pick some neuron $j$ in node $i$ on which the branching potentially leads to the largest improvement on the verified bounds:

$$\arg\max_{i,j} \min_{1 \leq k \leq K} \max_{\boldsymbol{\alpha}, \boldsymbol{\beta}} V(h, \underline{B}(\mathbf{l}, i, j, k), \overline{B}(\mathbf{u}, i, j, k), \boldsymbol{\alpha}, \boldsymbol{\beta}), \quad (10)$$

where we use $\underline{B}(\mathbf{l}, i, j, k)$ to denote the updated intermediate lower bounds for the $k$-th branch after branching neuron $j$ in node $i$, and similarly $\overline{B}(\mathbf{u}, i, j, k)$ for the upper bounds. Previous works typically use some branching heuristic (Bunel et al., 2018; 2020; Lu & Mudigonda, 2020; De Palma et al., 2021) which approximates the potential improvement of a branching in an efficient way.

Suppose we consider branching a neuron $j$ in node $i$ and we aim to estimate $V(\cdot)$ in Eq. (10) for each branch $k$. In linear bound propagation, when the bounds are propagated to node $i$, we have:

$$h(\mathbf{x}) \geq \mathbf{A}_{i,j}^{(k)} \hat{\mathbf{x}}_{i,j} + \mathbf{c}^{(k)} \geq V(h, \underline{B}(\mathbf{l}, i, j, k), \overline{B}(\mathbf{u}, i, j, k), \boldsymbol{\alpha}, \boldsymbol{\beta}), \quad (11)$$

where we use $\mathbf{A}_{i,j}^{(k)}$ and $\mathbf{c}^{(k)}$ to denote the parameters in the linear bounds for the $k$-th branch. Note that branching a neuron in node $i$ only affects the linear relaxation of nonlinear nodes immediately after node $i$ (i.e., output nodes of $i$), and thus $\mathbf{A}_{i,j}^{(k)}$ and $\mathbf{c}^{(k)}$ can be computed by only propagating the linear bounds from the output nodes of $i$ using stored linear bounds rather than from the ultimate output of $h(\mathbf{x})$. If we want to exactly obtain $V(h, \underline{B}(\mathbf{l}, i, j, k), \overline{B}(\mathbf{u}, i, j, k), \boldsymbol{\alpha}, \boldsymbol{\beta})$, then we need to further propagate the linear bounds until the input of the network, which is costly.

For a more efficient estimation, the BaBSR heuristic (Bunel et al., 2020) originally for ReLU networks essentially propagates the bounds only to the node before the branched one with an early stop, as they then ignore the coefficients ($\mathbf{A}_{i-1,j}^{(k)}$ for a feedforward NN) without propagating further. Note that we have described this heuristic in a general way, although it was originally for ReLU networks only. We call it "BaBSR-like" as a direct adaption from BaBSR (Bunel et al., 2020). However, we find a BaBSR-like branching heuristic is suboptimal on the models with general nonlinearities we experimented, as the heuristic ignores the important impact of the discarded coefficients on the verified bounds.

We propose a new branching heuristic named Branching via Bound Propagation with Short-cuts (BBPS), where we use a shortcut to directly propagate the bounds to the input. We expect it to more precisely estimate the potential improvement than simply discarding terms during the bound propagation, and more efficient than simply propagating the bounds layer by layer to the input. Specifically, we save the linear bounds of all the potentially branched intermediate layers during the initial verification before BaB. For every neuron $j$ in intermediate layer $i$, we record:

$$\forall \mathbf{x} \in \mathcal{C}, \quad \underline{\hat{\mathbf{A}}}_{ij}\mathbf{x} + \underline{\hat{\mathbf{c}}}_{ij} \leq \hat{\mathbf{x}}_{ij} \leq \overline{\hat{\mathbf{A}}}_{ij}\mathbf{x} + \overline{\hat{\mathbf{c}}}_{ij}, \tag{12}$$

where $\underline{\hat{\mathbf{A}}}_{ij}, \underline{\hat{\mathbf{c}}}_{ij}, \overline{\hat{\mathbf{A}}}_{ij}, \overline{\hat{\mathbf{c}}}_{ij}$ are parameters for the linear bounds. These are obtained when linear bound propagation is used for computing the intermediate bounds $[l_{i,j}, u_{i,j}]$ and the linear bounds are propagated to the input $\mathbf{x}$. We then use Eq. (12) to compute a lower bound for $\mathbf{A}_{i,j}^{(k)}\hat{\mathbf{x}}_{i,j} + \mathbf{c}^{(k)}$:

$$\forall \mathbf{x} \in \mathcal{C}, \quad \mathbf{A}_{i,j}^{(k)}\hat{\mathbf{x}}_{i,j} + \mathbf{c}^{(k)} \geq (\mathbf{A}_{i,j,+}^{(k)}\underline{\hat{\mathbf{A}}}_{ij} + \mathbf{A}_{i,j,-}^{(k)}\overline{\hat{\mathbf{A}}}_{ij})\mathbf{x} + \mathbf{A}_{i,j,+}^{(k)}\underline{\hat{\mathbf{c}}}_{ij} + \mathbf{A}_{i,j,-}^{(k)}\overline{\hat{\mathbf{c}}}_{ij} + \mathbf{c}^{(k)}, \tag{13}$$

and then the RHS can be concretized by Eq. (4) and serve as an approximation for $V(\cdot)$ after branching. In this way, the linear bounds are directly propagated from node $i$ to input $\mathbf{x}$ and concretized using a shortcut. Utilizing previously saved linear bounds has also been used in previous works (Shi et al., 2019; Zhong et al., 2021) for speeding up bound propagation, while we show that it can serve as a better branching heuristic for general nonlinearities as we will also empirically demonstrate.

## 4 EXPERIMENTS

**Settings.** We focus on verifying NNs with nonlineari­ties beyond ReLU which has been widely studied in prior works, and we experiment on models with various non­linearities as shown in Table 1. We mainly consider the commonly used $\ell_\infty$ robustness verification specification on image classification. We compare with baselines (Singh et al., 2019b; Müller et al., 2022c; Henriksen & Lomus­cio, 2020; Ryou et al., 2021; Bonaert et al., 2021; Wu et al., 2022; Wei et al., 2023) on models they support re­spectively. We adopt some MNIST (LeCun et al., 2010)

Table 1: List of models with various non-linearities in our experiments.

| Model | Nonlinearities in the model |
|---|---|
| Feedforward | sigmoid, tanh, sin, GeLU |
| LSTM | sigmoid, tanh, $xy$ |
| ViT with ReLU | ReLU, $xy$, $x/y$, $x^2$, $\sqrt{x}$, $\exp(x)$ |
| ML4ACOPF | ReLU, sigmoid, sin, $xy$, $x^2$ |

models from existing works (Singh et al., 2019a;b; Müller et al., 2022c), along with their data instances for verification. We also compute an upper bound on the number of potentially verifiable instances by PGD attack (Madry et al., 2018), as a sound verification should not verify on instances where a PGD attack can successfully discover counterexamples. Besides, we also train several new models on CIFAR-10 (Krizhevsky et al., 2009) by PGD adversarial training (Madry et al., 2018) using an $\ell_\infty$ perturbation with $\epsilon = 1/255$ in both training and verification. For these CIFAR-10 models, we first run vanilla CROWN (Zhang et al., 2020; Xu et al., 2020) (without $\alpha, \beta$ or BaB) and PGD attack (Madry et al., 2018) on the test set and remove instances on which either PGD attack succeeds or vanilla CROWN can already verify the property. Therefore, we only retain instances that can possibly be verified but are relatively hard to verify. If there are more than 100 instances after the filtering, we only retain the first 100 instances. We set a timeout of 300 seconds for our BaB in all these experiments. Details are in Appendix D. In addition, we also adopt an NN verification bench­mark for verifying properties in the Machine Learning for AC Optimal Power Flow (ML4ACOPF) problem, beyond robustness verification. And we show results on a ReLU network in Appendix C.2.

**Experiments on Sigmoid and Tanh networks for MNIST.** We first experiment on Sigmoid networks and Tanh networks. Table 2 shows the results. On 6 out of the 8 models, our BaB with BBPS is able

Table 2: Number of verified instances out of the first 100 test examples on MNIST for several Sigmoid networks and Tanh networks along with their $\epsilon$. The settings are the same as those in Müller et al. (2022c). "$L \times W$" in the network names denote a fully-connected NN with $L$ layers and $W$ hidden neurons in each layer. The upper bounds in the last row are computed by PGD attack.

| Method | Sigmoid Networks | | | | Tanh Networks | | | |
|---|---|---|---|---|---|---|---|---|
| | $6 \times 100$ $\epsilon = 0.015$ | $6 \times 200$ $\epsilon = 0.012$ | $9 \times 100$ $\epsilon = 0.015$ | ConvSmall $\epsilon = 0.014$ | $6 \times 100$ $\epsilon = 0.006$ | $6 \times 200$ $\epsilon = 0.002$ | $9 \times 100$ $\epsilon = 0.006$ | ConvSmall $\epsilon = 0.005$ |
| DeepPoly (Singh et al., 2019b)[a][b] | 30 | 43 | 38 | 30 | 38 | 39 | 18 | 16 |
| PRIMA (Müller et al., 2022c)[a] | 53 | 73 | 56 | 51 | 61 | 68 | 52 | 30 |
| VeriNet (Henriksen & Lomuscio, 2020)[c] | 65 | 81 | 56 | - | 31 | 30 | 16 | - |
| Wu et al. (2022)[?] | 65 | 75 | 96[?] | 63 | - | - | - | - |
| Vanilla CROWN (Zhang et al., 2018)[b] | 53 | 63 | 49 | 65 | 18 | 24 | 44 | 55 |
| $\alpha,\beta$-CROWN ($\alpha$ only w/o BaB) | 62 | 81 | **62** | 84 | **65** | 72 | 58 | 69 |
| Our BaB (BBPS) | **71** | **83** | **62** | **92** | **65** | **78** | **59** | **75** |
| Upper bound | 93 | 99 | 92 | 97 | 94 | 97 | 96 | 98 |

[a]Results for DeepPoly and PRIMA are directly from Müller et al. (2022c).
[b]While DeepPoly and CROWN are thought to be equivalent on ReLU networks (Müller et al., 2022c), these two works adopt different relaxation for Sigmoid and Tanh, which results in different results here.
[c]Results for VeriNet are obtained by running the tool (`https://github.com/vas-group-imperial/VeriNet`) by ourselves. VeriNet depends on the FICO Xpress commercial solver which requires a license for models that are relatively large. FICO Xpress declined the request we submitted for the academic license, directing us to obtain it via a (course) tutor, which is not applicable to our research. Thus results on ConvSmall models are not available.
[?]We found that the result Wu et al. (2022) reported on the Sigmoid $9 \times 100$ model exceeds the upper bound by PGD attack ($96 > 92$), and thus the result tends to be not fully valid. Results on Tanh networks are unavailable.

to verify additional instances over using $\alpha$ only and further boost the performance of verification, and our BaB outperforms all the non-CROWN baselines. We also find that improving on Sigmoid $9 \times 100$ and Tanh $6 \times 100$ networks by BaB is hard, as the initial bounds are typically too loose on the unverifiable instances, possibly due to these models being trained by standard training without robustness intervention in Müller et al. (2022c). In Figure 2, we plot the total number of verified instances against the running time for various methods, showing that our method can verify more instances compared to the baselines when the timeout threshold is at least around 10 seconds, and BaB enables us to verify more instances as more time is allowed compared to using $\alpha$ only. We also report the average running time in Appendix C.6.

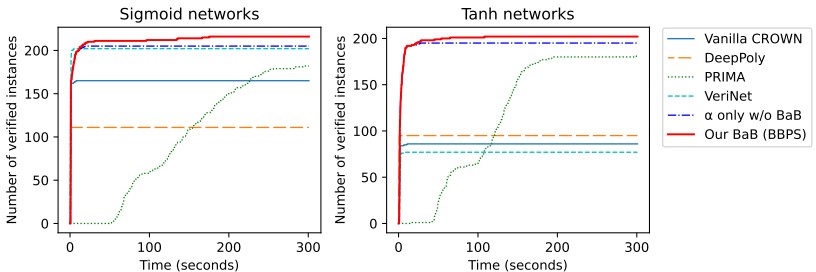

Figure 2: Total number of verified instances against running time threshold, on the three fully-connected Sigmoid networks (left) and three fully-connected Tanh networks (right) respectively in Table 2. The ConvSmall models are not included due to missing results for VeriNet.

**Experiments on feedforward networks with various activation functions on CIFAR-10.** In Table 3, we show results for models with various activation functions on CIFAR-10 trained by PGD. The results show that our BaB effectively improves verification beyond using $\alpha$ only without BaB. Besides, the ablation studies show that using our BBPS branching heuristic usually improves the performance over the BaBSR-like heuristic adapted from Bunel et al. (2020). Disabling $\beta$ optimization worsens the results, which validates the effectiveness of encoding the general branching constraints. For PRIMA and vanilla CROWN, as we only use relatively hard instances for verification here, these two methods are unable to verify any instance in this experiment. For VeriNet, all the models here are too large without a license for the FICO Xpress solver (we are unable to obtain an

Table 3: Number of verified instances out of 100 filtered instances on CIFAR-10 with $\epsilon = 1/255$ for feedforward networks with various activation functions.

| Method | Sigmoid Networks | | | | Tanh Networks | | Sine Networks | | | GeLU Networks | | |
|---|---|---|---|---|---|---|---|---|---|---|---|---|
| | 4×100 | 4×500 | 6×100 | 6×200 | 4×100 | 6×100 | 4×100 | 4×200 | 4×500 | 4×100 | 4×200 | 4×500 |
| PRIMA (Müller et al., 2022c)[a] | 0 | 0 | 0 | 0 | 0 | 0 | - | - | - | - | - | - |
| Vanilla CROWN[b] | 0 | 0 | 0 | 0 | 0 | 0 | 0 | 0 | 0 | 0 | 0 | 0 |
| $\alpha$ only w/o BaB[c] | 28 | 16 | 43 | 39 | 25 | 6 | 4 | 2 | 4 | 44 | 33 | 27 |
| BaB (BaBSR-like) | 34 | 17 | 44 | 41 | 35 | 8 | 54 | 30 | 10 | 64 | 53 | 37 |
| BaB (BBPS, w/o $\beta$) | 47 | 20 | 55 | 47 | 39 | 9 | 49 | 28 | 15 | 61 | 50 | 36 |
| Our BaB (BBPS) | **53** | **21** | **61** | **49** | **41** | **9** | **64** | **37** | **25** | **69** | **53** | **37** |

[a]Results for PRIMA are obtained by running ERAN (https://github.com/eth-sri/eran) which contains PRIMA. PRIMA does not support sine or GeLU activations.
[b]We have extended the support of vanilla CROWN to the GeLU activation, as discussed in Appendix B.3, which was not supported in the original code.
[c]For Sigmoid and Tanh networks, "$\alpha$ only w/o BaB" is equivalent to the existing $\alpha,\beta$-CROWN which has existing support for optimizable linear relaxation on Sigmoid and Tanh but not Sin or GeLU.

academic license as mentioned in Table 2); we have not obtained the code to run Wu et al. (2022) on these models either. Thus, we do not include the results for VeriNet or Wu et al. (2022).

**Experiments on LSTMs.** Next, we experiment on LSTMs containing more complex nonlinearities, including both Sigmoid and Tanh activations, as well as multiplication as $\text{sigmoid}(x)\tanh(y)$ and $\text{sigmoid}(x)y$. We compare with PROVER (Ryou et al., 2021) which is a specialized verification algorithm for RNN outperforming earlier RNN verification works (Ko et al., 2019). While there are other works on verifying RNN and LSTM, such as Du et al. (2021); Mohammadinejad et al. (2021); Paulsen & Wang (2022), we have not found their code, and we also make orthogonal contributions compared them on improving the relaxation for RNN verification. Thus, we omit them in our experiments. We take the hardest model, an LSTM for MNIST, from the main experiments of PROVER (other models can be verified by PROVER on more than 90% instances and are thus omitted), where each $28 \times 28$ image is sliced into 7 frames for LSTM. We also use two LSTMs trained by ourselves on CIFAR-10, where we linearly map each $32 \times 32$ image into 4 patches as the input tokens, similar to ViTs with patches (Dosovitskiy et al., 2021). Table 4 shows the results. Using $\alpha$ only without BaB can already outperform PROVER with specialized relaxation for RNN and LSTM, and using BaB further boosts the performance.

Table 4: Number of verified instances out of 100 instances on MNIST and CIFAR-10 LSTM networks. The MNIST model follows the setting of the hardest model in the main experiments of PROVER (Ryou et al., 2021) with $\epsilon = 0.01$. The CIFAR-10 models are trained by ourselves with $\epsilon = 1/255$. "LSTM-7-32" indicates an LSTM with 7 input frames and 32 hidden neurons, similar for the other two models. Results for PROVER are obtained by running the tool (https://github.com/eth-sri/prover).

| Method | MNIST Model (Ryou et al., 2021) | CIFAR-10 Models | |
|---|---|---|---|
| | LSTM-7-32 | LSTM-4-32 | LSTM-4-64 |
| PROVER (Ryou et al., 2021) | 63 | 8 | 3 |
| $\alpha$ only w/o BaB | 83 | 16 | 9 |
| BaB (BaBSR-like) | 84 | 17 | 12 |
| Our BaB (BBPS) | **86** | **25** | **15** |
| Upper bound | 98 | 100 | 100 |

Table 5: Number of verified instances on ViTs for CIFAR-10 ($\epsilon = 1/255$). "ViT-$L$-$H$" stands for $L$ layers and $H$ heads. For each model, there are fewer than 100 instances after the filtering, shown as the upper bounds. Results for DeepT are obtained by running the tool (https://github.com/eth-sri/DeepT).

| Method | ViT-1-3 | ViT-1-6 | ViT-2-3 | ViT-2-6 |
|---|---|---|---|---|
| DeepT (Bonaert et al., 2021) | 0 | 1 | 0 | 1 |
| $\alpha$ only w/o BaB | 1 | 3 | 11 | 7 |
| BaB (BaBSR-like) | 13 | 32 | 20 | 22 |
| Our BaB (BBPS) | **15** | **34** | **28** | **24** |
| Upper bound | 67 | 92 | 72 | 69 |

**Experiments on ViTs.** We also experiment on ViTs, which contain nonlinearities that are less studied as shown in Table 1. For ViTs, we compare with DeepT (Bonaert et al., 2021) which is specialized for verifying Transformers without using BaB. We show the results in Table 5, where our methods outperform DeepT and BaB effectively improves the verification. Besides, in Appendix C.1, we also compare with Wei et al. (2023) which supports verifying attention networks but not the entire ViT, and we experiment on models from Wei et al. (2023), where our methods also outperform Wei et al. (2023).

**Experiments on ML4ACOPF.** Finally, we experiment on models for the Machine Learning for AC Optimal Power Flow (ML4ACOPF) problem (Guha et al., 2019), and we adopt the ML4ACOPF neural network verification benchmark[2], a standardized benchmark in 2023 Verification of Neural Networks Competition (VNN-COMP). The benchmark consists of a NN with power demands as inputs, and the output of the NN gives an operation plan of electric power plants. Then, the benchmark aims to check for a few nonlinear constraint violations of this plan, such as power generation and balance constraints. These constraints, as part of the computational graph to verify, involve many nonlinearities including Sin, Sigmoid, multiplication, and square. Our framework is the first to support this verification problem. Among the 23 benchmark instances, PGD attack only succeeds on one instance, and our method (BaB + BBPS) verifies all the remaining 22 instances; without BaB, optimizing $\alpha$ only can verify only 16 instances in this benchmark.

## 5 RELATED WORK

Branch-and-bound (BaB) has been shown to be an effective technique for NN verification (Bunel et al., 2018; Lu & Mudigonda, 2020; Wang et al., 2018a; Xu et al., 2021; De Palma et al., 2021; Kouvaros & Lomuscio, 2021; Wang et al., 2021; Henriksen & Lomuscio, 2021; Shi et al., 2022), but most of the existing works focus on ReLU networks and are not directly applicable to networks with nonlinearities beyond ReLU. On BaB for NNs with other nonlinearities, Henriksen & Lomuscio (2020) conducted BaB on Sigmoid and Tanh networks, but their framework still depends on a commercial LP solver which has been argued as less effective than recent NN verification methods using linear bound propagation with branching constraints (Wang et al., 2021). Besides, Wu et al. (2022) studied verifying Sigmoid networks with counter-example-guided abstraction refinement, but their method is still specialized for Sigmoid. Moreover, these works have only considered S-shaped activations, and there lacks a general framework supporting general nonlinearities beyond some particular ones, which we address in this paper. Without using BaB, there are also other works studying the relaxation in verifying NNs with various nonlinearities, such as RNNs and LSTMs (Ko et al., 2019; Du et al., 2021; Ryou et al., 2021; Mohammadinejad et al., 2021; Zhang et al., 2023), and also Transformers (Shi et al., 2019; Bonaert et al., 2021; Wei et al., 2023). These works have orthogonal contributions compared to ours using BaB for further improvement above a base verifier. In addition, there are works studying the branching heuristic in verifying ReLU networks, such as filtering initial candidates with a more accurate computation (De Palma et al., 2021), using Graph Neural Networks for the heuristic (Lu & Mudigonda, 2020), or using a heuristic guided with tighter multiple-neuron relaxation (Ferrari et al., 2021), which may inspire future improvement on the BaB for general nonlinearities.

## 6 CONCLUSIONS

To conclude, we propose a general BaB framework for NN verification involving general nonlinearities. We also propose a new and more effective branching heuristic for BaB on general nonlinearities and we extend optimized linear relaxation. Experiments on verifying NNs with various nonlinearities demonstrate the effectiveness of our method.

**Limitations and Future work**. There remain several limitations in this work to be resolved in the future. As mentioned in Section 3.2, we have only used a simple way for deciding the branching points, and it will be interesting for future works to investigate more sophisticated ways. Besides, for the branching heuristic, future work may study the possibility of applying the latest progress on ReLU networks to strengthen the branching heuristic for general nonlinearities.

---

[2]https://github.com/AI4OPT/ml4acopf_benchmark

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

## A    ADDITIONAL ILLUSTRATION

We illustrate the overall framework in Figure 3.

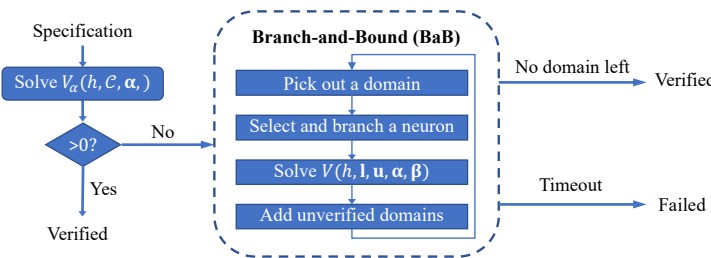

Figure 3: Illustration of our framework, as described in Section 3.1.

## B    ADDITIONAL OPTIMIZABLE LINEAR RELAXATION

In this section, we derive new optimizable linear relaxation for nonlinearities including multiplication, sine, and GeLU, which are not originally supported in $\alpha,\beta$-CROWN for optimizable linear relaxation.

### B.1    OPTIMIZABLE LINEAR RELAXATION FOR MULTIPLICATION

For each elementary multiplication $xy$ where $x \in [\underline{x}, \overline{x}]$, $y \in [\underline{y}, \overline{y}]$ are the intermediate bounds for $x$ and $y$, we aim to relax and bound $xy$ as:

$$\forall x \in [\underline{x}, \overline{x}], y \in [\underline{y}, \overline{y}], \quad \underline{a}x + \underline{b}y + \underline{c} \le xy \le \overline{a}x + \overline{b}y + \overline{c}, \tag{14}$$

where $\underline{a}, \underline{b}, \underline{c}, \overline{a}, \overline{b}, \overline{c}$ are parameters in the linear bounds. Shi et al. (2019) derived optimal parameters that minimize the gap between the relaxed upper bound and the relaxed lower bound:

$$\arg\min_{\underline{a}, \underline{b}, \underline{c}, \overline{a}, \overline{b}, \overline{c}} \int_{x \in [\underline{x}, \overline{x}]} \int_{y \in [\underline{y}, \overline{y}]} (\overline{a}x + \overline{b}y + \overline{c}) - (\underline{a}x + \underline{b}y + \underline{c}) \quad \text{s.t. Eq. (14).} \tag{15}$$

However, the optimal parameters they found only guarantee that the linear relaxation is optimal for this node, but not the final bounds after conducting a bound propagation on the entire NN. Therefore, we aim to make these parameters optimizable to tighten the final bounds as previous works did for ReLU networks or S-shaped activations (Xu et al., 2021; Lyu et al., 2020).

We notice that Shi et al. (2019) mentioned that there are two solutions for $\underline{a}, \underline{b}, \underline{c}$ and $\overline{a}, \overline{b}, \overline{c}$ respectively that solves Eq. (15):

$$\begin{cases} \underline{a}_1 = \underline{y} \\ \underline{b}_1 = \underline{x} \\ \underline{c}_1 = -\underline{x}\underline{y} \end{cases}, \quad \begin{cases} \overline{a}_1 = \overline{y} \\ \overline{b}_1 = \underline{x} \\ \overline{c}_1 = -\underline{x}\overline{y} \end{cases}, \tag{16}$$

$$\begin{cases} \underline{a}_2 = \overline{y} \\ \underline{b}_2 = \overline{x} \\ \underline{c}_2 = -\overline{x}\overline{y} \end{cases}, \quad \begin{cases} \overline{a}_2 = \underline{y} \\ \overline{b}_2 = \overline{x} \\ \overline{c}_2 = -\overline{x}\underline{y} \end{cases}. \tag{17}$$

Therefore, to make the parameters optimizable, we introduce parameters $\underline{\alpha}$ and $\overline{\alpha}$, and we interpolate between Eq. (16) and Eq. (17) as:

$$\begin{cases} \underline{a} = \underline{\alpha}\underline{y} + (1 - \underline{\alpha})\overline{y} \\ \underline{b} = \underline{\alpha}\underline{x} + (1 - \underline{\alpha})\overline{x} \\ \underline{c} = -\underline{\alpha}\underline{x}\underline{y} - (1 - \underline{\alpha})\overline{x}\overline{y} \end{cases} \quad \text{s.t. } 0 \le \underline{\alpha} \le 1, \tag{18}$$

$$\begin{cases} \overline{a} = \overline{\alpha}\overline{y} + (1 - \overline{\alpha})\underline{y} \\ \overline{b} = \overline{\alpha}\underline{x} + (1 - \overline{\alpha})\overline{x} \\ \overline{c} = -\overline{\alpha}\underline{x}\overline{y} - (1 - \overline{\alpha})\overline{x}\underline{y} \end{cases} \quad \text{s.t. } 0 \le \overline{\alpha} \le 1. \tag{19}$$

It is easy to verify that interpolating between two sound linear relaxations satisfying Eq. (14) still yields a sound linear relaxation. And $\underline{\alpha}$ and $\overline{\alpha}$ are part of all the optimizable linear relaxation parameters $\boldsymbol{\alpha}$ mentioned in Section 2.

## B.2 OPTIMIZABLE LINEAR RELAXATION FOR SINE

We also derive new optimized linear relaxation for periodic functions, in particular $\sin(x)$. For $sin(x)$ where $x \in [\underline{x}, \overline{x}]$, we aim to relax and bound $sin(x)$ as:

$$\forall x \in [\underline{x}, \overline{x}], \quad \underline{a}x + \underline{b} \leq \sin(x) \leq \overline{a}x + \overline{b}, \tag{20}$$

where $\underline{a}, \underline{b}, \overline{a}, \overline{b}$ are parameters in the linear bounds. A non-optimizable linear relaxation for $\sin$ already exists in $\alpha, \beta$-CROWN and we adopt it as an initialization and focus on making it optimizable. At initialization, we first check the line connecting $(\underline{x}, \sin(\underline{x}))$ and $(\overline{x}, \sin(\overline{x}))$, and this line is adopted as the lower bound or the upper bound without further optimization, if it is a sound bounding line.

Otherwise, a tangent line is used as the bounding line with the tangent point being optimized. Within $[\underline{x}, \overline{x}]$, if $\sin(x)$ happens to be monotonic with at most only one inflection point, the tangent point can be optimized in a way similar to bounding an S-shaped activation (Lyu et al., 2020), as illustrated in Figure 4.

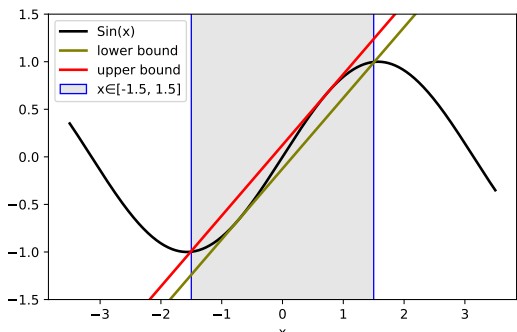

Figure 4: Linear relaxation for a Sin activation in an input range $[-1.5, 1.5]$ where the function is S-shaped.

Otherwise, there are multiple extreme points within the input range. Initially, we aim to find a tangent line that passes $(\underline{x}, \sin(\underline{x}))$ as the bounding line. Since $\underline{x}$ may be at different cycles of the sin function, we project into the cycle with range $[-0.5\pi, 1.5\pi]$, by taking $\underline{\tilde{x}}_l = \underline{x} - 2\underline{k}_l\pi$, where $\underline{k}_l = \lfloor \frac{\underline{x}+0.5\pi}{2\pi} \rfloor$. With a binary search, we find a tangent point $\underline{\alpha}_l$ on the projected cycle that satisfies

$$\sin'(\underline{\alpha}_l)(\underline{\alpha}_l - \underline{\tilde{x}}_l) + \sin(\underline{\tilde{x}}_l) = \sin(\underline{\alpha}_l), \tag{21}$$

which corresponds to a tangent point $\underline{t}_l = \underline{\alpha}_l + 2\underline{k}_l\pi$ at the original cycle of $\underline{x}$, and for any tangent point within the range of $[\underline{\alpha}_l + 2\underline{k}_l\pi, 1.5\pi + 2\underline{k}_l\pi]$, the tangent line is a valid lower bound. Similarly, we also consider the tangent line passing $(\overline{x}, \sin(\overline{x}))$, and we take $\overline{\tilde{x}}_l = \overline{x} - 2\overline{k}_l\pi$, where $\overline{k}_l = \lfloor \frac{\overline{x}-1.5\pi}{2\pi} \rfloor$, so that $\overline{\tilde{x}}_l$ is within range $[1.5\pi, 3.5\pi]$. We also conduct a binary search to find the tangent point $\overline{\alpha}_l$, which corresponds to to $\overline{\alpha}_l + 2\overline{k}_l\pi$ in the original cycle of $\overline{x}$, and for any tangent point within the range $[1.5\pi + 2\overline{k}_l\pi, \overline{\alpha}_l + 2\overline{k}_l\pi]$, the tangent line is also a valid lower bound. We make the tangent point optimizable with a parameter $\alpha_l$ $(\underline{\alpha}_l \leq \alpha_l \leq \overline{\alpha}_l)$, which corresponds to a tangent line at tangent point $t_l$ as the lower bound in Eq. (20) and Figure 5:

$$\begin{cases} \underline{a} = \sin'(t_l) \\ \underline{b} = \sin(t_l) - \underline{a}t_l \end{cases}, \quad \text{where} \quad \begin{cases} t_l = \alpha_l + 2\underline{k}_l\pi \ \text{ if } \ \underline{\alpha}_l \leq \alpha_l \leq 1.5\pi \\ t_l = \alpha_l + 2\overline{k}_l\pi \ \text{ if } \ 1.5\pi < \alpha_l \leq \overline{\alpha}_l \end{cases}. \tag{22}$$

In particular, when $\alpha_l = 1.5\pi$, both $\alpha_l + 2\underline{k}_l\pi$ and $\alpha_l + 2\overline{k}_l\pi$ are tangent points for the same tangent line.

The derivation for the upper bound is similar. We take $\underline{\tilde{x}}_u = \underline{x} - 2\underline{k}_u\pi$, where $\underline{k}_u = \lfloor \frac{\underline{x}-0.5\pi}{2\pi} \rfloor$, so that $\underline{\tilde{x}}_u$ is in range $[0.5\pi, 2.5\pi]$. And we take $\tilde{\overline{x}}_u = \overline{x} - 2\overline{k}_u\pi$, where $\overline{k}_u = \lfloor \frac{\overline{x}-2.5\pi}{2\pi} \rfloor$, so that

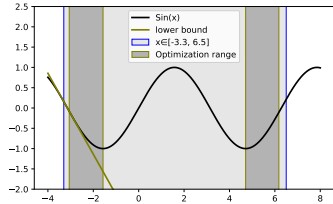 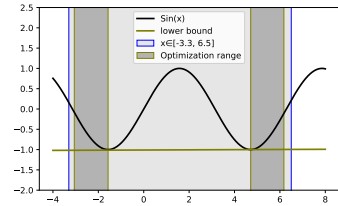 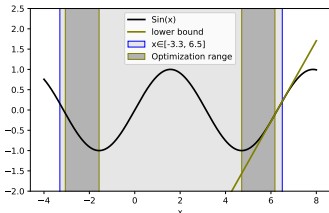

(a) The lower bound of Sin activation when $\alpha_l = \underline{\alpha}_l$.

(b) The lower bound of Sin activation when $\alpha_l = 1.5\pi$.

(c) The lower bound of Sin activation when $\alpha_l = \overline{\alpha}_l$.

Figure 5: Optimizing the lower bound of a Sin activation, where "Optimization range" shows all the valid tangent points for the lower bound during the optimization.

$\tilde{\underline{x}}_u$ is in range $[2.5\pi, 4.5\pi]$. Let $\underline{\alpha}_u$ be the tangent point where the tangent line crosses $\tilde{\underline{x}}_u$, and $\overline{\alpha}_u$ be the tangent point where the tangent line crosses $\tilde{\overline{x}}_u$, as found by a binary search. We define an optimizable parameter $\alpha_u$ ($\underline{\alpha}_u \le \alpha_u \le \overline{\alpha}_u$) which corresponds to a tangent line as the upper bound:

$$\begin{cases} \overline{a} = \sin'(t_u) \\ \overline{b} = \sin(t_u) - \overline{a}t_u \end{cases}, \quad \text{where} \quad \begin{cases} t_u = \alpha_u + 2\underline{k}_u\pi \text{ if } \underline{\alpha}_u \le \alpha_u \le 2.5\pi \\ t_u = \alpha_u + 2\overline{k}_u\pi \text{ if } 2.5\pi < \alpha_u \le \overline{\alpha}_u \end{cases}. \tag{23}$$

### B.3 Optimizable Linear Relaxation for GeLU

For GeLU function where $x \in [\underline{x}, \overline{x}]$ are the intermediate bounds for $x$, we aim to relax and bound $\text{GeLU}(x)$ as:

$$\forall x \in [\underline{x}, \overline{x}], \quad \underline{a}x + \underline{b} \le \text{GeLU}(x) \le \overline{a}x + \overline{b}, \tag{24}$$

where $\underline{a}, \underline{b}, \overline{a}, \overline{b}$ are parameters in the linear bounds.

Given input range $[\underline{x}, \overline{x}]$, if $\overline{x} \le 0$ or $\underline{x} \ge 0$, the range contains only one inflection point, the tangent point can be optimized in a way similar to bounding an S-shaped activation (Lyu et al., 2020). In other cases, $\underline{x} < 0$ and $\overline{x} > 0$ holds. For the upper bound, we use the line passing $(\underline{x}, \text{GeLU}(\underline{x}))$ and $(\overline{x}, \text{GeLU}(\overline{x}))$. For the lower bound, we derive two sets of tangent lines that crosses $(\underline{x}, \text{GeLU}(\underline{x}))$ and $(\overline{x}, \text{GeLU}(\overline{x}))$ with tangent points denoted as $\underline{\alpha}$ and $\overline{\alpha}$ respectively. We determine $\underline{\alpha}, \overline{\alpha}$ using a binary search that solves:

$$\begin{cases} \text{GeLU}'(\underline{\alpha})(\underline{\alpha} - \underline{x}) + \text{GeLU}(\underline{x}) = \text{GeLU}(\underline{\alpha}) \\ \text{GeLU}'(\overline{\alpha})(\overline{\alpha} - \overline{x}) + \text{GeLU}(\overline{x}) = \text{GeLU}(\overline{\alpha}) \end{cases}. \tag{25}$$

Any tangent line with a tangent point $\alpha$ ($\underline{\alpha} \le \alpha \le \overline{\alpha}$) is a valid lower bound, which corresponds to the lower bound in Eq. (24) with:

$$\begin{cases} \underline{a} = \text{GeLU}'(\alpha) \\ \underline{b} = \text{GeLU}(\alpha) - \alpha \text{GeLU}'(\alpha) \end{cases} \quad \text{s.t. } \underline{\alpha} \le \alpha \le \overline{\alpha}. \tag{26}$$

## C Additional Results

### C.1 Experiments on Self-Attention Networks from Wei et al. (2023)

To compare with Wei et al. (2023) that only supports verifying single-layer self-attention networks but not the entire ViT, we adopt pre-trained models from Wei et al. (2023) and run our verification methods under their settings, with 500 test images in MNIST using $\epsilon = 0.02$. We show the results in Table 6, where our methods also outperform Wei et al. (2023) on all the models.

### C.2 Experiments on a ReLU Network

In this section, we study the effect of our BBPS heuristic on ReLU activation. We adopt settings in Singh et al. (2019a;b); Müller et al. (2022c) and experiment on a "ConvSmall" model with ReLU

Table 6: Number of verified instances out of 500 instances in MNIST with $\epsilon = 0.02$. A-small, A-medium and A-big are three self-attention networks with different parameter sizes from Wei et al. (2023).

| Method | A-small | A-medium | A-big |
|---|---|---|---|
| Wei et al. (2023) | 406 | 358 | 206 |
| $\alpha$ only w/o BaB | 444 | 388 | 176 |
| BaB (BBPS) | **450** | **455** | **232** |
| Upper bound | 463 | 479 | 482 |

Table 7: Results on a "ConvSmall" model with ReLU activation (Singh et al., 2019a;b; Müller et al., 2022c) on 1000 instances from CIFAR-10. Percentage of instances verified by various methods are reported. For methods other than PRIMA, we use $\alpha,\beta$-CROWN as the underlying verifier but vary the branching heuristic. See explanation about the backup score in Appendix C.2.

| Method | Verified |
|---|---|
| PRIMA | 44.6% |
| BaBSR w/o backup score | 45.6% |
| BaBSR w/ backup score | 46.2% |
| Backup score only | 45.0% |
| BBPS w/o backup score | 46.0% |
| BBPS w/ backup score | 46.2% |

activation. The verification is evaluated on 1000 instances on CIFAR-10, following prior works. We show the results in Table 7, We find that on this ReLU network, our BBPS also works better than the BaBSR heuristic, when there is no *backup score* (46.0% verified by BBPS v.s. 45.6% verified by BaBSR). However, we find that recent works typically add a *backup score* for BaBSR, which is another heuristic score that serves as a backup for neurons with extremely small BaBSR scores. The backup score did not exist in the original BaBSR heuristic (Bunel et al., 2020) but it appeared in De Palma et al. (2021) and has also been adopted by works such as Wang et al. (2021) when using BaBSR for ReLU networks. This backup score basically uses the intercept of the linear relaxation for the upper bound of a ReLU neuron that needs branching. Unlike BaBSR or BBPS, the backup score does not aim to directly estimate the change on the bounds computed by bound propagation, but aims to use the intercept to reflect the reduction of the linear relaxation after the branching. When the backup score is combined with BaBSR or BBPS for ReLU networks, the backup score seems to dominate the performance, where both BaBSR and BBPS have similar performance with the backup score added (46.2% verified), which hides the underlying improvement of BBPS over BaBSR by providing a more precise estimation. However, the backup score is specifically for ReLU, assuming that the intercept of the linear relaxation can reflect the reduction of the linear relaxation, which is not the case for general nonlinearities. We leave it for future work to study the possibility of designing a backup score for general nonlinearities.

## C.3 EXPERIMENTS ON THE NUMBER OF BRANCHES

Table 8: Number of verified instances out of 100 filtered test examples on CIFAR-10 with $\epsilon = 1/255$, for our BaB with different number of branches $K$. The models and the verified specifications follow Table 3.

| Method | Sigmoid Networks | | | | Sine Networks | | |
|---|---|---|---|---|---|---|---|
| | $4 \times 100$ | $4 \times 500$ | $6 \times 100$ | $4 \times 200$ | $4 \times 100$ | $4 \times 200$ | $4 \times 500$ |
| Our BaB ($K = 2$) | 54 | 20 | 62 | 49 | 54 | 34 | 20 |
| Our BaB ($K = 3$) | 53 | 21 | 61 | 49 | **63** | **39** | **25** |

As mentioned in Section 3.2, we mainly use $K = 3$ for the number of branches in our main experiments, to demonstrate the ability of our framework for handling a general number of branches. In this section, we conduct a study on the impact of $K$. In Table 8, we compare the performance of our BaB with $K = 2$ and $K = 3$, respectively. On Sigmoid networks, we find that $K = 2$ and $K = 3$ yield comparable results. However, on networks with the $\sin$ activation which is more nonlinear compared to Sigmoid, using $K = 3$ significantly outperforms $K = 2$. This result is also consistent with the illustration in Figure 1b and 1c, where using 3 branches yields much tighter linear relaxation than using 2 branches for the $\sin$ activation. The results demonstrate the effectiveness and potential of our framework supporting a general number of branches, on NNs involving functions that are relatively more nonlinear.

## C.4 Experiments on Filtering in the Branching Heuristic

De Palma et al. (2021) proposed a branching heuristic named Filtered Smart Branching (FSB) which filters initial candidates selected by estimated bound improvements, with an additional step to compute more precise bound improvements by full linear bound propagation. The filtering mechanism is independent from our contribution on BBPS for computing initial estimations. In this section, we study the effect of the filtering mechanism when it is combined with the BaBSR-like heuristic and our BBPS heuristic, respectively, for selecting the initial candidates. Table 9 shows the results. We find that the filtering does not improve the performance here, possibly because their improved bound estimation is not sufficiently strong compared to the additional cost. Adding filtering sometimes hurts the performance more for BBPS compared to BaBSR-like, as BBPS tends to verify more hard instances, where the additional cost tends to cause timeout more easily when the runtime is originally long for the hard instances. We leave it for future work to study improving the filtering mechanism for NNs with general nonlinearities.

Table 9: Number of verified instances out of 100 filtered test examples on CIFAR-10 with $\epsilon = 1/255$, for BaB with the BaBSR-like heuristic and our BBPS, respectively, when the filtering in the branching heuristic, is enabled and disabled, respecitvely. The models and the verified specifications follow Table 3.

| Heuristic | Filtering | Sigmoid Networks | | | Tanh Networks | | |
|---|---|---|---|---|---|---|---|
| | | $4 \times 100$ | $4 \times 500$ | $6 \times 100$ | $4 \times 200$ | $4 \times 100$ | $6 \times 100$ |
| BaBSR-like | $\times$ | 34 | 17 | 44 | 41 | 35 | 8 |
| BaBSR-like | $\checkmark$ | 33 | 17 | 44 | 41 | 34 | 8 |
| BBPS | $\times$ | 53 | 21 | 61 | 49 | 41 | 9 |
| BBPS | $\checkmark$ | 50 | 21 | 56 | 48 | 41 | 9 |

## C.5 Experiments on Models with Other Robust Training Algorithms

We also experiment on models trained by robust training algorithms other than PGD. As a case study, we consider Small Adversarial Bounding Regions (SABR) (Müller et al., 2022b). SABR propagates adversarially selected small boxes for certified robust training and it relies on a complete verifier at test time. With SABR, we train 4 fully-connected Sigmoid networks on CIFAR-10 with $\epsilon = 1/255$, and we show the results in Table 10. The results demonstrate that our BaB with the BBPS heuristic effectively verifies more hard instances, compared to the verifier without BaB or BaB with the BaBSR-like heuristic.

## C.6 Results on the Running Time

In Figure 2, we have plotted the total number of verified instances against various running time threshold. In this section, we report the average running time for the experiment in Figure 2, and we show the results in Table 11. We first compute the average running time only on instances verified by each method. The average running time of our BaB is slightly higher than most baselines and is much lower than PRIMA. This evaluation tends to bias towards methods that solve much fewer hard instances, as hard instances tend to require larger running time. Therefore, we also compute the average running time on all the instances, where we use the timeout (300 seconds in our experiments)

Table 10: Number of verified instances out of the 100 filtered test examples on CIFAR-10 for several Sigmoid networks trained by SABR with $\epsilon = 1/255$.

| Method | Sigmoid Networks | | | |
| --- | --- | --- | --- | --- |
| | $4\times100$ | $4\times500$ | $6\times100$ | $6\times200$ |
| Vanilla CROWN | 0 | 0 | 0 | 0 |
| $\alpha$ only w/o BaB | 59 | 51 | 60 | 65 |
| BaB (BaBSR-like) | 73 | 65 | 71 | 76 |
| Our BaB (BBPS) | **74** | **75** | **76** | **77** |

as the running time for the instances that are not verified. Under this evaluation, the average running time of our BaB is lower than all the other methods. However, this evaluation can still be affected by the timeout and is thus not perfect either. Compared to the average running time, the plots in Figure 2 can more comprehensively reflect the time cost and the number of verified instances of different methods.

Table 11: Average running time of different methods for models in Figure 2.

| Method | Average running time (s) | | | |
| --- | --- | --- | --- | --- |
| | Verified instances | | All instances | |
| | Sigmoid Networks | Tanh Networks | Sigmoid Networks | Tanh Networks |
| Vanilla CROWN | 0.91 | 1.34 | 135.50 | 214.39 |
| DeepPoly | 0.66 | 0.46 | 189.24 | 205.15 |
| PRIMA | 139.55 | 107.84 | 202.12 | 184.06 |
| VeriNet | 0.68 | 1.81 | 98.45 | 223.46 |
| $\alpha,\beta$-CROWN ($\alpha$ only w/o BaB) | 1.78 | 2.87 | 96.22 | 106.87 |
| Our BaB (BBPS) | 5.48 | 4.47 | **87.94** | **101.01** |

## D  IMPLEMENTATION DETAILS

**Verification.**  We implement our verification algorithm based on auto_LiRPA[3] and $\alpha,\beta$-CROWN[4], both under the BSD-3-Clause license. We use the Adam optimizer (Kingma & Ba, 2015) to optimize $\boldsymbol{\alpha}$ and $\boldsymbol{\beta}$ with an initial learning rate of 0.1 and the learning rate is decayed by 2% after each iteration. To solve $V_\alpha(h, \mathcal{C}, \boldsymbol{\alpha})$ in the initial verification, we optimize $\boldsymbol{\alpha}$ for at most 100 iterations. And to solve $V(h, \mathbf{l}, \mathbf{u}, \boldsymbol{\alpha}, \boldsymbol{\beta})$ during BaB, we optimize $\boldsymbol{\alpha}$ and $\boldsymbol{\beta}$ for at most 50 iterations. Our BaB is batched where multiple domains are branched in parallel, and the batch size is dynamic tuned based on the model size to fit the GPU memory.

**Training the models.**  To train our models on CIFAR-10, we use PGD adversarial training (Madry et al., 2018). We use 7 PGD steps during the training and the step size is set to $\epsilon/4$. For training the Sigmoid networks in Table 3, we use the SGD optimizer with a learning rate of $5 \times 10^{-2}$ for 100 epochs; and for training the Tanh networks, we use the SGD optimizer with a learning rate of $1 \times 10^{-2}$ for 100 epochs. For training the LSTMs in Table 4, we use the Adam optimizer with a learning of $10^{-3}$ for 30 epochs. And for training the ViTs, we use the Adam optimizer with a learning of $5 \times 10^{-3}$ for 100 epochs. For Sin networks, we use the SGD optimizer with a learning rate of $1 \times 10^{-3}$ for 100 epochs

---

[3]`https://github.com/Verified-Intelligence/auto_LiRPA`
[4]`https://github.com/Verified-Intelligence/alpha-beta-CROWN`

