# OpenReview forum: "Formal Verification for Neural Networks with General Nonlinearities via Branch-and-Bound"
_ICLR.cc/2024/Conference — ICLR 2024 Conference Withdrawn Submission_

### Official Review · Reviewer_abyN · 2023-10-26

**Soundness:** 4 excellent
**Presentation:** 3 good
**Contribution:** 2 fair
**Rating:** 5
**Confidence:** 4

**Summary:**

This work extends the linear-bound-propagation and BaB based neural network verification framework $\alpha\beta$-CROWN to support branching on any node of the computational graph and thus on inputs to arbitrary non-linearities. Technically, these additional constraints are enforced using Lagrange multipliers. Additionally, a novel branching heuristic, BBPS, is introduced that leverages precomputed linear bounds as "shortcuts" to compute better approximations of the branching effect. The effectiveness of the method is demonstrated on a wide variety of activation functions.

**Strengths:**

* The tackled issue of (certified) adversarial robustness is of high importance.
* To the best of my knowledge, this paper is the first to describe the application of the popular BaB paradigm to general non-linearities.
* The paper combines well established techniques (general cutting planes and optimisable relaxation slopes) to enable branching for general non-linearities.
* The novel branching heuristic is elegant and effective (for non-ReLU activations).
* The extensive empirical evaluation is convincing and clearly shows improved performance over a wide range of baselines.

**Weaknesses:**

* Prior work and novel contributions are not always distinguished clearly, i.e., it is not immediately clear that Section 3.1 describes the prior work $\alpha\beta$-CROWN. Similarly, it is not clear that the general branching constraints described in Section 3.3 constitute special cases of the General Cutting Planes described by Zhang et al. (2022).
* The technical contribution beyond BBPS, seem limited as both enforcing arbitrary constraints on intermediate activations (Zhang et al. 2022) and optimising relaxation parametrisation was already possible in the  $\alpha\beta$-CROWN version this work builds on.


**References**
Zhang, Huan, et al. "General cutting planes for bound-propagation-based neural network verification."  NeurIPS 2022

**Questions:**

### Questions
1) Can you give an intuition on why BBPS does not seem to yield any improvement on ReLU networks while being crucial for other non-linearities?
2) At the bottom of page 5, you state that branching a neuron in node i only affects the linear relaxations of nonlinear nodes immediately after node i. Can tighter bounds there not lead to tighter bounds in later nodes and thus changing relaxations?
3) What is the impact of the branching factor $K$ on the resulting precision? Can you ablate its effect on one of the CIFAR10 networks, where branching was particularly effective? Can the same neuron be split multiple times in your framework?

### Conclusion
The authors successfully establish the effectiveness of BaB for general non-linearities and the advantages of their novel branching heuristic on a wide range of benchmarks and compared to a diverse set of baselines. While the technical novelty seems limited, I believe that demonstrating the applicability and effectiveness of established techniques in this setting is a valuable contribution in itself and am thus leaning to accept the paper. However, I believe the authors should make sure that novel contributions (Section 3.2 and 3.4) are clearly distinguished from prior work (Section 3.1 and 3.3), and have thus reduced my score. I am happy to raise it once this concern is addressed.

---

> ### Author Response · Authors · 2023-11-21
> **Response to Reviewer abyN**
>
> We thank the reviewer for the review and identifying the merits of our paper. Following the insightful feedback by the reviewer, we have revised the paper to more clearly distinguish our contributions from prior works. We also respond to the questions.
>
> ## Distinguishing contributions
>
> >Prior work and novel contributions are not always distinguished clearly, i.e., it is not immediately clear that Section 3.1 describes the prior work α,β-CROWN. Similarly, it is not clear that the general branching constraints described in Section 3.3 constitute special cases of the General Cutting Planes described by Zhang et al. (2022).
>
> We have added a paragraph in the beginning of Section 3.1 to clearly mention that the overall framework follows α,β-CROWN. Compared to α,β-CROWN, our methodological  contributions are on a general branching framework for general nonlinearities and also a new branching heuristic, which are discussed in the remaining subsections of Section 3.
>
> We have also revised Section 3.3 to clarify the relation and difference between our contribution and Zhang et al., 2022. Our general branching constraints are indeed a particular type of cutting plane constraints that can be utilized by Zhang et al., 2022 to tighten bounds in linear bound propagation. However, our contribution is on formulating general branching constraints, while Zhang et al., 2022 is mainly on utilizing general  constraints once the constraints are already formulated.
>
> ## Improvement on ReLU networks
>
> >Can you give an intuition on why BBPS does not seem to yield any improvement on ReLU networks while being crucial for other non-linearities?
>
> We study this problem in Appendix C.2. Basically, BaBSR in De Palma et al. (2021) has a backup score which dominates the performance on the ReLU networks we experimented, when the backup score is combined with BaBSR or BBPS. When the backup score is not used, BBPS still improves over BaBSR (Table 7). The backup score is specifically designed for ReLU networks, and we leave it for future work to study the possibility of designing backup scores for general nonlinearities. More details are in Appendix C.2.
>
> ## Intermediate bounds after a branching
>
> >At the bottom of page 5, you state that branching a neuron in node i only affects the linear relaxations of nonlinear nodes immediately after node i. Can tighter bounds there not lead to tighter bounds in later nodes and thus changing relaxations?
>
> We fix the intermediate bounds except for neurons that are branched. Indeed branching can potentially also tighten the intermediate bounds of later nodes. However, re-computing intermediate bounds can be costly as the number of subdomains after branching can be quite large. We only re-compute the final output bounds following α,β-CROWN (see Section B.1 of Wang et al., 2021).
>
> ## Number of branches
>
> >What is the impact of the branching factor K on the resulting precision? Can you ablate its effect on one of the CIFAR10 networks, where branching was particularly effective?
>
> We mainly used $K=3$ for the number of branches in our main experiments, to demonstrate the ability of our framework for handling a general number of branches. We have added new results in Appendix C.3 to compare the performance of $K=2$ and $K=3$.
>
> | |  Sigmoid 4x100 | Sigmoid 4x500 | Sigmoid 6x100 | Sigmoid 6x200 |
> | --- | --- | --- | --- | --- |
> | $K=2$ | 54 | 20 | 62 | 49 |
> | $K=3$ | 53 | 21 | 61 | 49 |
>
> | | Sin 4x100 | Sin 4x200 | Sin 4x500|
> | --- | --- | --- | --- |
> | $K=2$ | 54 | 34 | 20 |
> | $K=3$ | 63 | 39 | 25 |
>
> On Sigmoid networks, we find that $K=2$ and $K=3$ yield comparable results. However, on networks with the $\sin$ activation which is more nonlinear compared to Sigmoid, using $K=3$ significantly outperforms $K=2$. The results demonstrate the effectiveness and potential of our framework supporting a general number of branches, on NNs involving functions that are relatively more nonlinear.
>
> >Can the same neuron be split multiple times in your framework?
>
> Yes, the same neuron may be split multiple times if it is selected by the branching heuristic multiple times.

---

> > ### Comment · Reviewer_abyN · 2023-11-22
> > **Thanks for the response**
> >
> > I want to thank the authors for answering my questions.
> >
> > However, my concerns regarding the novelty of the proposed method remain. In particular, the fact that prior work already applied BaB techniques to S-shaped activations (Henriksen and Lomuscio, 2020) should have been made much clearer. Generalizing from S-shaped to general activation functions seems to mostly be an implementation rather than a methodological challenge.
> >
> > Given that branching constraints for more than two branches at a time are a special case of the general cutting planes discussed by Zhang et al. (2022), it seems like the only novelty of this work is the improved branching heuristic. While it is empirically effective, I believe this contribution falls short of the standard for a publication at ICLR and can thus not recommend acceptance.

---

> ### Author Response · Authors · 2023-11-23
> **Thank you for your prompt response and I hope the reviewer can reconsider our contribution**
>
> We greatly appreciate your prompt response!
>
> We acknowledge that (Henriksen and Lomuscio, 2020) conducted an initial study with a similar idea in a restricted setting. However, our approach is general, thorough, principled, and systematic, and we also conducted comprehensive experiments to demonstrate the power of our approach. We achieved state-of-the-art results on many new settings (including Transformer) and also enabled new applications of NN verification in ACOPF, which could never have happened without our efforts.
>
> Similarly, (Zhang et al. 2022) also did not consider branch and bound at all - although we share some techniques (e.g., Lagrangian duality), (Zhang et al. 2022) never demonstrated the effectiveness of branch-and-bound on general nonlinear functions, and in fact their paper used purely ReLU networks. We believe that **using well-established mathematical techniques in a new setting and demonstrating impressive empirical results** are sufficient contributions for a conference paper.
>
> **We believe the contributions of a paper are not restricted to narrowly defined “completely new ideas”**, and we hope the reviewer can consider our contribution to the field of NN verification in the long term. Many existing works are restrictive to ReLU or simple networks (like feedforward S-shape activation) and are roadblocks to making any practical impact. Our systematic extension of powerful BaB-based verification techniques will enable many novel applications for the verification of general computation graph, and has enabled completely new settings like ACOPF. None of the existing work like (Henriksen and Lomuscio, 2020) and (Zhang et al. 2022) can enable these.
>
> Thank you again for your constructive feedback, and we sincerely hope you can reevaluate our contribution.

---

### Official Review · Reviewer_NV4g · 2023-10-28

**Soundness:** 3 good
**Presentation:** 2 fair
**Contribution:** 2 fair
**Rating:** 5
**Confidence:** 4

**Summary:**

The paper presents a generalisation of a very popular network verifier, $\alpha$-$\beta$-CROWN, to new nonlinearities. In particular, branching support for sigmod, tanh, sine, GeLU and multiplication is added, along with bounding support (in terms of optimizable linear bounds) for the last three. Results show that the proposed techniques are more effective than relevant baselines on the considered settings.

**Strengths:**

The work extends the support of a state-of-the-art network verifier to nonlinearities beyond piece-wise linear.
As one would expect, the resulting framework remains effective on the considered non-linearities. In particular, the authors show that the presented framework works reasonably well on LSTM (on vision data) and ViTs.

**Weaknesses:**

While the work is definitely of interest to the practitioners in the area, the vast majority of the presented material is a fairly straightforward extension of very well known concepts in the literature. It would not appear to me that the extensions presented great technical challenges that needed to be overcome. Indeed, while branch-and-bound is mostly employed on piece-wise nonlinearities in the context of neural network verification, it is a fairly general concept which the authors simply instantiated on more variants of the neural network verification problem.

More in detail, the support of more nonlinearities in the bounding phase is a trivial applications of concepts presented in (Zhang et al., 2018; Xu et al., 2020). And, in practice, the $\alpha$-$\beta$-CROWN already extended the concept of optimizable linear bound propagation beyond ReLU (through support for sigmoid and tanh). As a result, extending these ideas to more nonlinearities is quite incremental. Similarly, previous work has already extended activation splitting to non-ReLU activations (Henriksen and Lomuscio, 2020). I believe this incrementality should be acknowledged more in the motivational sections of the paper.
While the branching part could have more room for improvement, the authors focus on extending a relatively old branching heuristic (BaBSR) that is typically preferred by more effective strategies in state-of-the-art works (FSB in $\beta$-CROWN, and the custom strategy introduced in MN-BaB). Furthermore, the authors do not justify the use of ternary branching (k=3) with uniform spacing between the branching points.

The experimental results are also on mostly toy problems and with fairly small perturbation sizes (1/255 as opposed to 2/255 and 8/255 typically employed in the literature on CIFAR-10). The authors sometimes select the properties to verify by excluding the properties that would be verified by CROWN (for instance, table 3). Taking the above into account, I am not sure how significant some of the improvements with respect to pre-existing work are (for instance, $\alpha$-$\beta$-CROWN without branching on Figure 2 and table 3).

In conclusion, I believe that most of the merit of the work pertains to the implementation. I am not sure this meets the bar for an ICLR publication.

**Questions:**

- Could the authors justify their choice for ternary (and uniform) branching? For instance, an ablation study on the number of branching points would be useful.

- I found the explanation of the branching strategy to be quite confusing. For instance, the presentation of BaBSR heavily differs from the one from the original authors, which is based on computing coefficients that estimate the impact of splitting on the last layer bounds from the (Wong and Kolter 2018) paper. Could the authors explain why the two presentations are equivalent?

---

> ### Author Response · Authors · 2023-11-21
> **Response to Reviewer NV4g (1/2)**
>
> We thank the reviewer for the review. We have revised the paper to acknowledge previous works more clearly, and we have also added results on the impact of different numbers of branches. We also respond to the concerns on the applicability of other branching heuristics, the problem complexity in the experiments, and the evaluation. Finally, we also explain the formulation for BaBSR.
>
> ## Acknowledging previous works more
>
> >And, in practice, the α,β-CROWN already extended the concept of optimizable linear bound propagation beyond ReLU (through support for sigmoid and tanh). As a result, extending these ideas to more nonlinearities is quite incremental. Similarly, previous work has already extended activation splitting to non-ReLU activations (Henriksen and Lomuscio, 2020). I believe this incrementality should be acknowledged more in the motivational sections of the paper.
>
> We have revised Section 1 to acknowledge the previous works more. We have mentioned that α,β-CROWN does support non-ReLU activations but their BaB is still restricted to ReLU. We have also extended the discussion on the previous works about BaB for S-shaped activations including Henriksen and Lomuscio, 2020.
>
> ## Branching heuristic
>
> >While the branching part could have more room for improvement, the authors focus on extending a relatively old branching heuristic (BaBSR) that is typically preferred by more effective strategies in state-of-the-art works (FSB in α,β-CROWN, and the custom strategy introduced in MN-BaB).
>
> Our contribution on the branching heuristic is focused on providing a more precise estimation on the bound improvement for neural networks with general nonlinearities. FSB and the heuristic in MN-BaB still need to first estimate the bound improvement for all the neurons.
>
> FSB is based on BaBSR, and it further has a filtering mechanism, which is independent from our contribution on BBPS, as filtering can be combined with generic branching heuristics. We experiment with "BaBSR-like + filtering" and "BBPS+filtering":
>
> | |  Sigmoid 4x100 | Sigmoid 4x500 | Sigmoid 6x100 | Sigmoid 6x200 | Tanh 4x100 | Tanh 6x100 |
> | --- | --- | --- | --- | --- | --- | --- |
> | BaBSR-like | 34 | 17 | 44 | 41 | 35 | 8 |
> | BaBSR-like + filtering | 33 | 17 | 44 | 41 | 34 | 8 |
> | BBPS | 53 | 21 | 61 | 49 | 41 | 9 |
> | BBPS + filtering | 50 | 21 | 56 | 48 | 41 | 9 |
>
> We find that the filtering does not improve the performance on these models, possibly because their improved bound estimation is not sufficiently strong compared to the additional cost. Adding filtering sometimes hurts the performance more for BBPS compared to BaBSR-like, as BBPS tends to verify more hard instances, where the additional cost tends to cause timeout more easily when the runtime is originally long for the hard instances. We leave it for future work to study improving the filtering mechanism for NNs with general nonlinearities. We have revised the paper, and we have added the results and discussions Appendix C.4.
>
> MN-BaB proposed Active Constraint Score Branching (ACS). ACS specifically considers multi-neuron relaxation for providing a better estimation, when multi-neuron relaxation is used for ReLU activations. This is not applicable in our case because we do not have multi-neuron relaxation. We have not added multi-neuron relaxation, because in Table 2, we find that α,β-CROWN without BaB (which does not have multi-neuron relaxation) can already outperform PRIMA which has multi-neuron relaxation, on the Sigmoid/Tanh networks. However, it will be interesting for future works to further consider multi-neuron relaxation and make it more effective on general nonlinearities.
>
> In MN-BaB's branching strategy, there is also Cost Adjusted Branching which considers the cost of re-computing the intermediate bounds, which is also not applicable here. We fix the intermediate bounds except for neurons that are branched, to save the cost of re-computing the intermediate bounds. This follows α,β-CROWN (mentioned in Appendix B.1 of Wang et al., 2021).

---

> ### Author Response · Authors · 2023-11-21
> **Response to Reviewer NV4g (2/2)**
>
> ## Number of branches
>
> >Could the authors justify their choice for ternary (and uniform) branching? For instance, an ablation study on the number of branching points would be useful.
>
> We mainly used $K=3$ for the number of branches in our main experiments, to demonstrate the ability of our framework for handling a general number of branches. We have added new results in Appendix C.3 to compare the performance of $K=2$ and $K=3$.
>
> | |  Sigmoid 4x100 | Sigmoid 4x500 | Sigmoid 6x100 | Sigmoid 6x200 |
> | --- | --- | --- | --- | --- |
> | $K=2$ | 54 | 20 | 62 | 49 |
> | $K=3$ | 53 | 21 | 61 | 49 |
>
> | | Sin 4x100 | Sin 4x200 | Sin 4x500|
> | --- | --- | --- | --- |
> | $K=2$ | 54 | 34 | 20 |
> | $K=3$ | 63 | 39 | 25 |
>
> On Sigmoid networks, we find that $K=2$ and $K=3$ yield comparable results. However, on networks with the $\sin$ activation which is more nonlinear compared to Sigmoid, using $K=3$ significantly outperforms $K=2$. The results demonstrate the effectiveness and potential of our framework supporting a general number of branches, on NNs involving functions that are relatively more nonlinear. So far, we have only used uniform branching points, for its simplicity, but it will be interesting for future works to study the possibility of optimizing branching points for stronger performance.
>
> ## Problem complexity
>
> >The experimental results are also on mostly toy problems and with fairly small perturbation sizes (1/255 as opposed to 2/255 and 8/255 typically employed in the literature on CIFAR-10).
>
> Verifying networks with general nonlinearities is inherently more challenging than verifying networks with ReLU activation only. On non-ReLU networks, previous works on NN verification also used relatively small perturbations. For example, on CIFAR-10, Henriksen and Lomuscio, 2020 used $\epsilon\in\{ 0.05/255,0.1/255,0.2/255,0.5/255,1/255 \}$ (Table 2 in Henriksen and Lomuscio, 2020). On MNIST, PRIMA (Müller et al., 2022b) used $\epsilon$ between 0.002 and 0.015 (Table 5 in Müller et al., 2022b), which is also much smaller than 0.3 or 0.4 often used in adversarial robustness literature on MNIST. Moreover, we not only have "toy problems", but also have models for the practical ​Machine Learning for AC Optimal Power Flow (ML4ACOPF) problem.
>
> ## Filtering instances to verify
>
> >The authors sometimes select the properties to verify by excluding the properties that would be verified by CROWN (for instance, table 3).  Taking the above into account, I am not sure how significant some of the improvements with respect to pre-existing work are (for instance, α,β-CROWN without branching on Figure 2 and table 3).
>
> We focus on solving hard instances (instances that cannot be solved by vanilla CROWN in Zhang et al., 2018) in the verification problem. Our empirical results reflect how many of the hard instances each method can solve and demonstrate the significant effectiveness of our framework on solving the hard instances (e.g., in Table 3, our BaB is able to solve 53% of the hard instances on the 4x100 Sigmoid network, while α,β-CROWN without branching is only able to solve 28% of the hard instances). Easy instances can be verified by most of the recent methods anyway, but the percentage of hard instances can vary for different settings, depending on the difficulty of the verification problem. Thus, we believe it makes more sense to only evaluate on the hard instances, to benchmark the effectiveness of the recent verification methods proposed after the vanilla CROWN.
>
> ## Explanation on the branching strategy
>
> >I found the explanation of the branching strategy to be quite confusing. For instance, the presentation of BaBSR heavily differs from the one from the original authors, which is based on computing coefficients that estimate the impact of splitting on the last layer bounds from the (Wong and Kolter 2018) paper. Could the authors explain why the two presentations are equivalent?
>
> When the linear relaxation of the activation functions is the same, Wong and Kolter, 2018 and CROWN (Zhang et al., 2018) are equivalent (Salman et al., 2019), and they are just derived from two different views (primal view in Zhang et al., 2018 v.s. dual view in Wong and Kolter, 2018) with different linear relaxation in their implementation. Given this equivalence, the coefficients from Wong and Kolter, 2018 utilized by the original BaBSR formulation correspond to the coefficients in CROWN's linear bounds ($A$ in Eq. (1)) which we use in this paper. The reviewer may refer to Salman et al., 2019 for more explanation on the equivalence and unification of the convex relaxation-based NN verification methods.
>
> Salman, H., Yang, G., Zhang, H., Hsieh, C. J., & Zhang, P. (2019). A convex relaxation barrier to tight robustness verification of neural networks. Advances in Neural Information Processing Systems, 32.

---

> > ### Comment · Reviewer_NV4g · 2023-12-04
> > **Thank you for your response**
> >
> > I thank the authors for their response.
> > I am glad that the authors are willing to more clearly acknowledge previous work. However, also taking the discussion with other reviewers into account, I still believe that the main contributions of the submission are fairly incremental and mostly linked to the implementation, and hence fall below the publication threshold.
> >
> > I am confident that this work could become stronger if it was to include some of the following:
> > - A stronger branching heuristic, for instance including the selection of the branching point(s): given that BBPS is the main technical contribution, it is a pity that the work lacks a detailed study over K (what about other values >3?) and non-uniform branching points.
> > - A set of benchmarks making a stronger case for BaB-based verification on non-piecewise-linear networks. I would think ML4ACOPF to be fairly niche, and I am still not convinced by the vision benchmarks, which I called "toy problems" because they fail to motivate why one would want to use (and, as a result, verify) the employed architectures on MNIST/CIFAR-10, especially if the final goal is to attain high certified accuracy.
> >
> > Concerning the branching strategy: I am aware of the correspondence (when the relaxation uses the same linear bounds) between the presentation from CROWN and the one from the (Wong and Kolter 2018) paper. While the paper would definitely benefit from an explicit explanation of this correspondence (I think a reader comparing the presentation in (Bunel et al., 2020) with the one from the authors would be fairly puzzled), my own confusion arises from the fact that the original BaBSR computes the branching scores as differences of terms pertaining to ambiguous and non-ambiguous neurons arising in the output bound computations (see equation (9) in (Bunel et al., 2020)). I fail to see a correspondence between those scores and the presentation in section 3.4, which, as far as I can understand, relies on an approximation of the backward propagation of the split bounds.

---

### Official Review · Reviewer_ygGn · 2023-10-31

**Soundness:** 3 good
**Presentation:** 3 good
**Contribution:** 2 fair
**Rating:** 5
**Confidence:** 4

**Summary:**

The paper extends the α,β-Crown verification framework to support the
verification of neural networks with general activation functions. In
particular it introduces a novel branching mechanism that allows for splitting
a neuron in more than two branches. It additionally presents a variant of the
BaBSR branching heuristic for selecting the neuron to split at each step. The
experimental results reported show improvements over the state-of-the-art
verifiers on some common and on some more complex benchmarks.

**Strengths:**

Novel extension of the α,β-Crown framework to tackle general activation
functions. Good experimental evaluation showing the efficacy of the resulting
method.

**Weaknesses:**

- Highly incremental to α,β-Crown - the resulting method is essentially
  α,β-Crown with support for more than two branches per split.

- The BBPS branching heuristic a trivial variant of BaBSR, it gives marginal
  gains, and it is not compared with other preciser variants of BaBSR from  De
  Palma et al., 2021.

**Questions:**

Please see comments above.

---

> ### Author Response · Authors · 2023-11-21
> **Response to Reviewer ygGn**
>
> We thank the reviewer for the review. We respond to the concerns on the comparison to α,β-CROWN and the existing branching heuristics below:
>
> ## Comparison to α,β-CROWN
>
> While this work is based on α,β-CROWN, we make important contributions on supporting general nonlinearities, as it will enable researchers from various domains (such as power flow as we demonstrate in the paper) to verify their non-ReLU networks. We generalize the verification framework to support BaB for general nonlinearities, which is not only about supporting more than two branches, but also others including general and non-zero branching points. We also propose a new branching heuristic crucial for general nonlinearities. These are all new methodological contributions compared to α,β-CROWN.
>
> ## BBPS
>
> >The BBPS branching heuristic a trivial variant of BaBSR, it gives marginal gains
>
> We believe our new BBPS heuristic is not a "trivial variant of BaBSR". First, BaBSR was originally formulated for ReLU only, while BBPS is generally formulated for general nonlinearities, allowing for BaB on networks with general nonlinearities. We also propose to utilize previously saved linear bounds (Eq. (13) in the revised paper) to make more precise estimation, which is also new compared to BBPS which simply ignored the term we estimated. Empirically, our BBPS archives significant gains over BaBSR in most cases. For example., in Table 3, BBPS verifies 53 instances while BaBSR verifies 34 instances for the Sigmoid 4x100 network, out of 100 instances, which is a +55.9% relative gain. Thus, we think the gains from BBPS is not marginal.
>
> >it is not compared with other preciser variants of BaBSR from De Palma et al., 2021
>
> The FSB heuristic in De Palma et al., 2021 has a filtering mechanism which runs a full bound propagation for shortlisted candidates. Filtering is independent from our contribution on BBPS for computing initial estimations, as filtering can be combined with generic branching heuristics. We experiment with "BaBSR-like + filtering" and "BBPS+filtering":
>
> | |  Sigmoid 4x100 | Sigmoid 4x500 | Sigmoid 6x100 | Sigmoid 6x200 | Tanh 4x100 | Tanh 6x100 |
> | --- | --- | --- | --- | --- | --- | --- |
> | BaBSR-like | 34 | 17 | 44 | 41 | 35 | 8 |
> | BaBSR-like + filtering | 33 | 17 | 44 | 41 | 34 | 8 |
> | BBPS | 53 | 21 | 61 | 49 | 41 | 9 |
> | BBPS + filtering | 50 | 21 | 56 | 48 | 41 | 9 |
>
> We find that the filtering does not improve the performance on these models, possibly because their improved bound estimation is not sufficiently strong compared to the additional cost. Adding filtering sometimes hurts the performance more for BBPS compared to BaBSR-like, as BBPS tends to verify more hard instances, where the additional cost tends to cause timeout more easily when the runtime is originally long for the hard instances. We leave it for future work to study improving the filtering mechanism for NNs with general nonlinearities. We have revised the paper, and we have added the results and discussions in Appendix C.4.

---

### Official Review · Reviewer_Cfzz · 2023-11-01

**Soundness:** 2 fair
**Presentation:** 3 good
**Contribution:** 3 good
**Rating:** 6
**Confidence:** 3

**Summary:**

The paper introduces a verification framework with BaB for neural networks that 1) encompasses general branching points and an arbitrary number of branches, which could generalize NN verification to variety of networks with various activation functions; 2) develops a novel branching heuristic named BBPS with a more accurate estimation; 3) enables verification on models for the ACOPF application.

**Strengths:**

1. BBPS constantly outperforming existing SOTA neural network verification on several benchmark datasets.
2. The author conducts experiments on different network architectures, it's interesting to see the study about effectiveness of neural network verification on modern architectures like ViT.
3. The paper is well written and easy to follow.

**Weaknesses:**

1. It seems that all the classifiers are trained on PGD, it would be better if authors could report a more comprehensive evaluation on other robust training algorithms.
2. It would be better if the author could provide a comparison including quantitative results of average running time for clarity.

**Questions:**

Please refer to the questions in Weaknesses.

---

> ### Author Response · Authors · 2023-11-21
> **Response to Reviewer Cfzz**
>
> We thank the reviewer for the review. We have revised the paper to address the weakness points mentioned by the reviewer. We also respond to the questions below:
>
> ## Other robust training algorithms
>
> We thank the reviewer for the suggestion on including models with robust training techniques other than PGD. We added an experiment on models trained by Small Adversarial Bounding Regions (SABR) (Mueller et al., 2022) which is a method for certified training. We trained 4 Sigmoid networks with SABR and evaluated the performance of verification. We show the results in the table below. Our BaB with the BBPS heuristic verifies 67 more instances in total, compared to the method without BaB, and it also verifies 17 more instances compared to BaB with the BaBSR-like heuristic. The results demonstrate that our BaB with the BBPS heuristic effectively verifies more hard instances. We have also added the results and more details in Appendix C.5.
>
> |  | Sigmoid 4x100 | Sigmoid 4x500 | Sigmoid 6x100 | Sigmoid 6x200 |
> | --- | --- | --- | --- | --- |
> | $\alpha$ only w/o BaB | 59 | 51 | 60 | 65 |
> | BaB (BaBSR-like) | 73 | 65 | 71 | 76 |
> | Our BaB (BBPS) | 74 | 75 | 76 | 77 |
>
> Mueller, M. N., Eckert, F., Fischer, M., & Vechev, M. (2022, September). Certified Training: Small Boxes are All You Need. In The Eleventh International Conference on Learning Representations.
>
> ## Running time
>
> We have added results on the average running time in Appendix C.6, for the setting in Figure 2.
>
> Average running time computed on verified instances:
> | | Sigmoid Networks | Tanh Networks |
> | --- | --- | --- |
> | Vanilla CROWN | 0.91 | 1.34 |
> | DeepPoly | 0.66 | 0.46 |
> | PRIMA | 139.55 | 107.84 |
> | VeriNet | 0.68 | 1.81 |
> | α,β-CROWN (α only w/o BaB) | 1.78 | 2.87 |
> | Our BaB (BBPS) | 5.48 | 4.47 |
>
> We first compute the average running time only on instances verified by each method. The average running time of our BaB is slightly higher than most baselines and is much lower than PRIMA. This evaluation tends to bias towards methods that solve much fewer hard instances, as hard instances tend to require larger running time.
>
> Average running time computed on all instances:
> | | Sigmoid Networks | Tanh Networks |
> | --- | --- | --- |
> | Vanilla CROWN | 135.50 | 214.39 |
> | DeepPoly | 189.24 | 205.15 |
> | PRIMA | 202.12 | 184.06 |
> | VeriNet | 98.45 | 223.46 |
> | α,β-Crown (α only w/o BaB) | 96.22 | 106.87 |
> | Our BaB (BBPS) | 87.94 | 101.01 |
>
> Therefore, we also compute the average running time on all instances, where we use the timeout (300 seconds in our experiments) as the running time for the instances that are not verified. Under this evaluation, the average running time of our BaB is lower than all the other methods. However, this evaluation is still not perfect as it can be affected by the timeout value.
>
> Compared to the average running time, we think that the plots in Figure 2, which show the numbers of instances against various running time thresholds, can more comprehensively reflect the time cost to verify different numbers of instances by each method.